# Chronology-based architecture of descending circuits that underlie the development of locomotor repertoire after birth

**Avinash Pujala, Minoru Koyama***

Janelia Research Campus, Howard Hughes Medical Institute, Ashburn, United States

**Abstract** The emergence of new and increasingly sophisticated behaviors after birth is accompanied by dramatic increase of newly established synaptic connections in the nervous system. Little is known, however, of how nascent connections are organized to support such new behaviors alongside existing ones. To understand this, in the larval zebrafish we examined the development of spinal pathways from hindbrain V2a neurons and the role of these pathways in the development of locomotion. We found that new projections are continually layered laterally to existing neuropil, and give rise to distinct pathways that function in parallel to existing pathways. Across these chronologically layered pathways, the connectivity patterns and biophysical properties vary systematically to support a behavioral repertoire with a wide range of kinematics and dynamics. Such layering of new parallel circuits equipped with systematically changing properties may be central to the postnatal diversification and increasing sophistication of an animal's behavioral repertoire.

DOI: https://doi.org/10.7554/eLife.42135.001

*For correspondence:
koyamam@janelia.hhmi.org

**Competing interests:** The authors declare that no competing interests exist.

## Introduction

At the time of birth, most animals are only capable of a limited set of reflexive behaviors that ensure their immediate survival. However, as development progresses, their behavioral repertoire rapidly diversifies and they become capable of producing increasingly refined and cognitive behaviors (*Kagan and Herschkowitz, 2006*; *Harlow and Harlow, 1965*; *Fox, 1965*; *Drapeau et al., 2002*). Concurrent with such changes, many new connections are rapidly being formed in the nervous system (*Gilmore et al., 2018*); *Semple et al. (2013)*; *Levitt, 2003*), suggesting that the formation of new connections is linked to the development of the behavioral repertoire. However, despite many studies that carried out circuit-level examination of the postnatal formation of new connections (*Morrie and Feller, 2016*; *Polley et al., 2013*; *Stein and Stanford, 2013*; *Kano and Watanabe, 2013*) very little is known still of how nascent connections are organized to support a behavioral repertoire that grows by including increasingly more sophisticated behaviors while retaining vital reflexive behaviors.

Recent studies in developing zebrafish have shed some light on the neural underpinnings of behavior development. As in most animals, the behavioral repertoire of zebrafish quickly diversifies after birth (hatching) to include more sophisticated behaviors (*Drapeau et al., 2002*; *McLean and Fetcho, 2009*). At the time of birth, a zebrafish is mostly quiescent but in response to a tactile stimulus exhibits crude locomotor behaviors that entail powerful and large bends of the body such as seen during escape and struggle (*Figure 1A*, Escape, Struggle) (*McLean and Fetcho, 2009*; *Liao and Fetcho, 2008*). This indicates that both of these early-born behaviors require strong

**eLife digest** Newborn babies have limited abilities. Indeed, most of our actions shortly after birth are the result of reflexes that serve our most basic need: to stay alive. As we get older, however, our behaviour gradually becomes more sophisticated. During this time, the billions of cells in our brain form new connections to build intricate 'circuits' of neurons that allow for more complicated thoughts and actions.

It is clear that the brain circuits that support new behaviours must develop in a way that does not interfere with the existing circuits that are vital for survival. However, the challenge has been to find a way to peer into a brain as it develops to see how these new circuits form.

In recent years, zebrafish have revolutionised research into neuronal circuits in animals. Developing over the course of a few days, these small transparent fish provide a window into the brain during the earliest stages of development. Indeed, the circuits of neurons that descend from the brain and connect to the spinal cord have already been mapped in these animals. Now, Pujala and Koyama have begun to follow the careful development of these 'descending' neurons, and relate it to the appearance of new behaviours in young zebrafish.

Time-lapse imaging with a fluorescent protein that is active only in specific descending neurons revealed that new circuits are laid down over existing ones, like the growth rings in a tree. Next, at different timepoints in zebrafish development, Pujala and Koyama traced these neurons backwards from the spine to the brain to identify which connections formed first. This showed that the spinal connections develop one after the other, in the same order that the neurons mature.

Next, Pujala and Koyama asked how the activity of neurons that mature early or late in development relates to specific behaviours in young zebrafish. Early-born circuits connect to neurons that produce powerful, reflex-driven, whole-body movements such as an escape response. The later circuits connect to different neurons through slower, less direct pathways; the late-born neurons also generate the refined movements that are acquired later in a zebrafish's development and help the fish to explore its environment.

These findings show that descending circuits in zebrafish run parallel to each other, but with distinct connections and properties that allow them to control different kinds of movements. While this study was conducted using an animal model, a better understanding of how such circuits develop and the movements they control may one day aid the treatment of patients with neurodegenerative diseases or injuries where connections have been lost.

DOI: https://doi.org/10.7554/eLife.42135.002

activation of axial muscles across a large portion of the fish's body despite their differences with respect to the the direction of motion and tail beat frequency (escape is forward locomotion with fast tail beat frequency while struggle is backward locomotion with slow tail beat frequency). However, by 2 days after birth, the fish also spontaneously exhibit a more refined locomotor behavior wherein only the caudal portion of the tail moves (*Figure 1A*, Spontaneous swim) with relatively weaker and slower wave-like undulating movements (*McLean and Fetcho, 2009*, *Figure 1A*; *Mirat et al., 2013*). Such movements are indicative of slow propagation of moderate axial muscle activity from the rostral to the caudal end of the moving part of the tail. With this type of tail-restricted locomotion, the fish is able to move forward by keeping its head, and therefore gaze, stable. This ability may be critical for the subsequent development of visually guided behaviors such as prey capture. Escape, one of the crude locomotor patterns that emerge early after the birth of a zebrafish, is controlled by spinal interneurons that are born well before the animal's birth (*McLean and Fetcho, 2009*; *Kimura et al., 2006*; *Eklöf-Ljunggren et al., 2012*), whereas the more refined and weaker locomotion pattern that emerges after an animal's birth is controlled by spinal interneurons that are born around the time of an animal's birth (*McLean et al., 2007*; *Satou et al., 2012*; *McLean and Fetcho, 2009*). This indicates that the sequential emergence of new and more sophisticated locomotor patterns during development is mediated at the level of the spinal cord by the addition of distinct functional groups of neurons rather than by fine control of a single functional group (*Fetcho and McLean, 2010*). Similarly in the mammalian spinal cord, it has been shown more recently that distinct functional groups emerge in sequence during development (*Tripodi et al.,*

*2011*). This finding has led to the idea that even in mammals, the development of the locomotor repertoire is supported by the addition of new functional groups (*Tripodi and Arber, 2012*). However, it has yet to be revealed how the connections from the brain to emerging spinal functional groups are established so that new and existing spinal groups can be recruited appropriately by the brain to support the diversification of the locomotor repertoire.

Locomotion-related signals from the brain reach the spinal cord by way of excitatory reticulospinal neurons of the hindbrain (*Dubuc et al., 2008*; *Grillner and Georgopoulos, 1996*; *Jordan et al., 2008*; *Roberts et al., 2008*). V2a neurons in the hindbrain, similar to the ones in the spinal cord, are glutamatergic and project their axons ipsilaterally (*Cepeda-Nieto et al., 2005*; *Kinkhabwala et al., 2011*; *Kimura et al., 2013*; *Bouvier et al., 2015*). They include excitatory reticulospinal neurons (*Cepeda-Nieto et al., 2005*; *Kimura et al., 2013*; *Bouvier et al., 2015*) and are capable of initiating

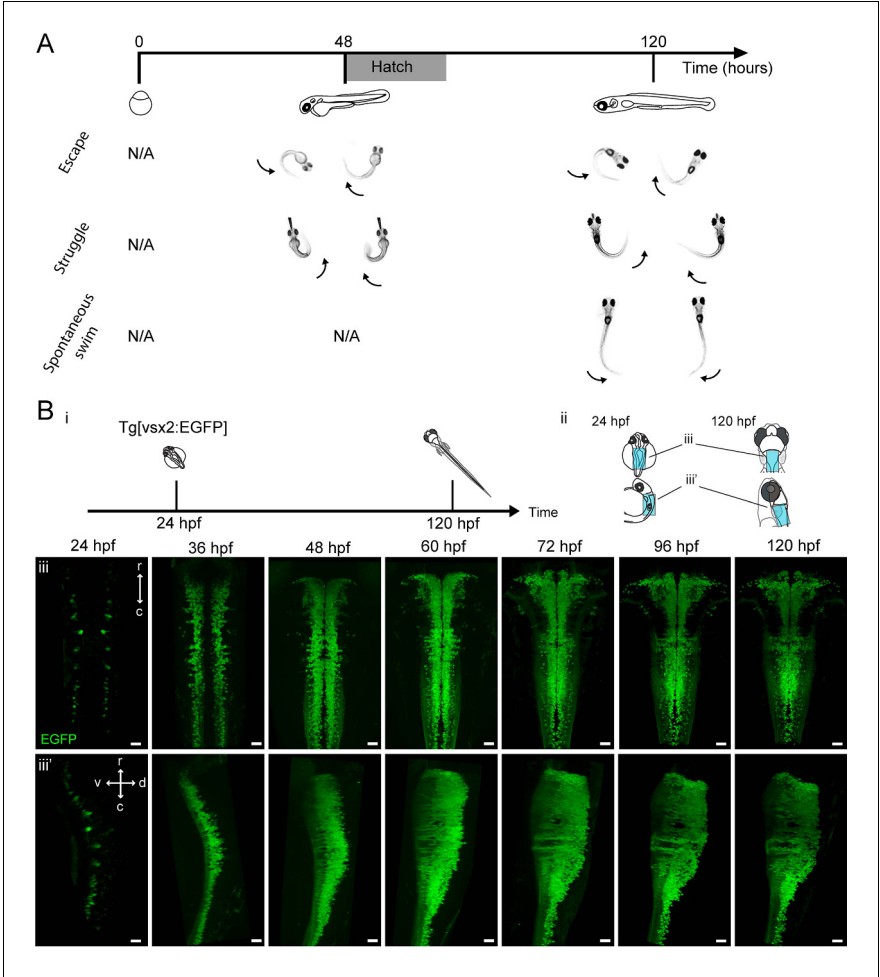

**Figure 1.** Development of zebrafish locomotor behaviors and genesis of hindbrain V2a neurons. (**A**) Development of locomotor behaviors in zebrafish. By the time of birth (48–72 hpf), in response to strong stimuli, zebrafish are capable of generating escapes and struggles that are characterized by strong whole body bends. During these behaviors fish can assume a C-shaped body posture as a result of simultaneous activation of ipsilateral axial muscles spanning most body segments (arrows). By 120 hpf, a fish also spontaneously exhibits weaker swimming (Spontaneous swim), which is characterized by bending of mostly the caudal portion of the tail by a relatively small amount. This indicates that the axial muscles in the caudal part of the tail are activated relatively weakly. (**B**) Emergence of hindbrain V2a neurons expressing EGFP. (**i**) Timing of experiments. (**ii**) Regions displayed (cyan patch) in subsequent panels. (**iii**) Dorsal and side views of hindbrain V2a neurons from 24 hpf to 120 hpf. The caudal end of the volume roughly corresponds to the boundary between hindbrain and spinal cord. r, rostral; c, caudal; d, dorsal; v, ventral. Scale bars, 30 µm.
DOI: https://doi.org/10.7554/eLife.42135.003

and terminating locomotion (*Kimura et al., 2013*; *Bouvier et al., 2015*). Interestingly, the hindbrain V2a neurons in larval zebrafish show a topographical organization that is related to differentiation time such that newly-born neurons stack dorsally to pre-existing ones (*Kinkhabwala et al., 2011*). Furthermore, in a subset of the hindbrain V2a neurons, it has been shown that the ventral, and therefore early-born, population gets recruited during escape-like locomotion with fast tail beat frequency while the dorsal, and therefore late-born, population gets recruited during spontaneous swim-like locomotion with slow tail beat frequency (*Kinkhabwala et al., 2011*). Even though it is unknown whether this finding holds true for the entire hindbrain V2a population, it suggests that, even in the hindbrain V2a neurons, distinct functional groups emerge in sequence and contribute to the development of locomotor behaviors. Such hindbrain organization raises the possibility that a new hindbrain functional group connects selectively to an age-matched spinal functional group to produce a novel locomotion pattern. However, it is also possible that hindbrain functional groups emerging in succession provide increasingly finer excitatory drive to all spinal groups, but each hindbrain group recruits only the age-matched spinal group because of the higher excitability of the latter compared to its predecessors (*McLean et al., 2007*). Indeed, in the case of midbrain descending pathways, the recruitment of spinal neurons is determined not by selectivity of the descending inputs these neurons receive but by the biophysical properties they express (*Wang and McLean, 2014*). Thus, it remains to be resolved how the the hindbrain descending neurons establish connections to the spinal functional groups during development to support the postnatal diversification and increased sophistication of locomotor patterns.

Here, we examined the development of spinal pathways from the hindbrain V2a neurons and the role of these pathways in the development of locomotion. We show that early-born V2a descending neurons that project to the spinal cord early in development are only recruited during the stimulus-elicited crude and powerful locomotor patterns that appear early in development whereas their late-born counterparts that project to the spinal cord late in development are recruited during the more refined and weaker spontaneous locomotion that appears later in development. Moreover, the spinal projections of these two populations form spatially-distinct neuropil layers and give rise to parallel pathways instead of forming a series of non-selective pathways that differ in the strengths of their connections. Furthermore, these parallel pathways differ in their connectivity patterns and biophysical properties in a manner suitable for the locomotor behavior they participate in. The early-born group makes direct connections to a class of motoneurons capable of producing strong axial muscle activity across a large extent of the spinal cord. The early-born group also expresses biophysical properties with fast time constants. Both these features are well-suited for producing the crude types of locomotion that appear early in development in that they support the near simultaneous activation of axial muscles across a large extent of the fish's body. On the other hand, the late-born group connects to the late-born spinal interneurons that are active during refined locomotion and express biophysical properties with slow kinetics. These interneurons innervate to a class of motoneurons that produce weaker axial muscle activity in the caudal spinal cord. These features make the late-born group more suitable for producing the refined locomotion that appears later in development in that they support the moderate axial muscle activity in the caudal portion of the tail. Indeed, ablation of each group produced deficits in distinct motor patterns: ablation of the early-born group weakened the sensory-elicited crude and fast locomotion whereas ablation of the late-born group affected the more refined and slower spontaneous locomotion. Altogether, we reveal in the descending circuits a chronologically-layered parallel architecture underlying the diversification and increased sophistication of motor patterns in larval zebrafish. Even though chronotopic neuropil organization similar to what we have described here has been observed in many neural systems (*Espinosa and Luo, 2008*; *Tripodi et al., 2011*; *Voigt et al., 1993*; *Walsh and Guillery, 1985*; *Kulkarni et al., 2016*; *Brierley et al., 2009*; *Brierley et al., 2012*), it has not been demonstrated in these systems how such organization relates to the postnatal development of behaviors. The findings we report here suggest that the chronological layering of parallel circuits and the systematic variation in the associated connectivity patterns and biophysical properties are fundamental attributes of the nervous system that form the basis for an animal's ability to exhibit new and increasingly sophisticated behaviors after birth while maintaining vital reflexive behaviors.

## Results

### Neuronal birth order dictates cell body position and order of spinal projections in hindbrain V2a population

To link the development of the locomotor repertoire to the development of hindbrain V2a neurons and their spinal pathways, we systematically examined the ontogeny of these neurons, their topography and the onset of their projections to the spinal cord until 120 hr post-fertilization (hpf) when larvae are capable of exhibiting both stimulus-evoked crude locomotion that appears by 48 to 72 hpf and more refined spontaneous locomotion that appears by 96 to 120 hpf (*Figure 1A*) (*Drapeau et al., 2002*; *McLean and Fetcho, 2009*).

We first examined the ontogeny of hindbrain V2a neurons from 24 hr post fertilization (hpf) to 120 hpf with time lapse imaging (*Figure 1B*, n = 8 for each timepoint) of a transgenic line expressing EGFP under the control of the promoter of *vsx2*, a transcription factor specific to V2a neurons. This transcription factor has been shown to become active after the final cell division (*Kimura et al., 2008*). Therefore, the onset of EGFP expression in a V2a neuron in our transgenic line indicates the time of differentiation - which we use synonymously with the time of birth - of this neuron. At 24 hpf, one or two pairs of V2a neurons appeared in a segmental fashion (*Figure 1B* iii, iii', 24 hpf), and then the number of V2a neurons increased dramatically until 60 hpf (*Figure 1B* iii, iii', 60 hpf). After 72 hpf, the V2a cluster showed no drastic visible changes in the overall shape (*Figure 1B* iii, iii', 72–120 hpf). As a fish is still only capable of generating crude forms of locomotion at 72 hpf, this suggests that the connectivity of its neurons and their electrophysiological properties need to mature to support refined locomotion.

We then examined where V2a neurons born at different time points resided in the hindbrains of 120 hpf fish that are capable of generating both the crude and refined forms of locomotion. We did this by photoconverting Kaede, a fluorescent photoconvertible protein, which was expressed under the control of the *vsx2* promoter. We converted Kaede throughout the fish at different time points (24–96 hpf) in different groups of fish and imaged each of these groups at 120 hpf (*Figure 2—figure supplement 1A* i, n = 8 for each timepoint). Based on the presence of converted Kaede (shown in magenta), we distinguished neurons that already had Kaede at the time of photoconversion from those that started to express Kaede after the time of photoconversion (*Kimura et al., 2006*; *Caron et al., 2008*). In most rhombomeres, V2a neurons born by 24 hpf were located in the most lateral portions of the brain region occupied by this group of neurons (*Figure 2—figure supplements 1C* and 24 hpf conversion, magenta cells), whereas neurons born afterwards gradually filled up the space medial to the pre-existing ones (*Figure 2—figure supplement 1C* and 36–72 hpf conversion). Compared to photoconversion at earlier time points, photoconversion at 72 hpf revealed relatively few, if any, unconverted green cells, except in the rostral hindbrain (*Figure 2—figure supplements 1C* and 72 hpf conversion). This indicated that the neurogenesis of hindbrain V2a neurons was mostly complete by 72 hpf when fish are still only capable of producing the crude locomotion, suggesting that further developments of these neurons are required if they are to contribute to the development of locomotor behaviors.

Hindbrain V2a neurons with spinal projections (*Video 1*) are more likely to play roles in the development of the locomotor repertoire, therefore we examined the times of birth of these neurons in further detail. First, we focused on large spinal projecting neurons (reticulospinal neurons)

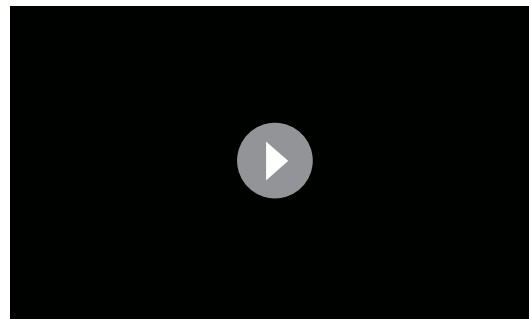

**Video 1.** Hindbrain V2a neurons with descending projections. A series of horizontal slices showing hindbrain V2a neurons with descending processes (magenta). These neurons were labeled by photoconversion of Kaede in the rostral spinal cord (muscle segments 5–7) at 108 hpf. The confocal stack was acquired at five dpf and then registered to Zebrafish Brain Browser (ZBB) atlas. Key cell groups are highlighted as the horizontal slice moves from dorsal to ventral. Magenta, photoconverted Kaede; green, unconverted Kaede; gray, brain structure volume. Magenta, photoconverted Kaede; green, unconverted Kaede; gray, brain structure volume. Rostral to the left.
DOI: https://doi.org/10.7554/eLife.42135.006

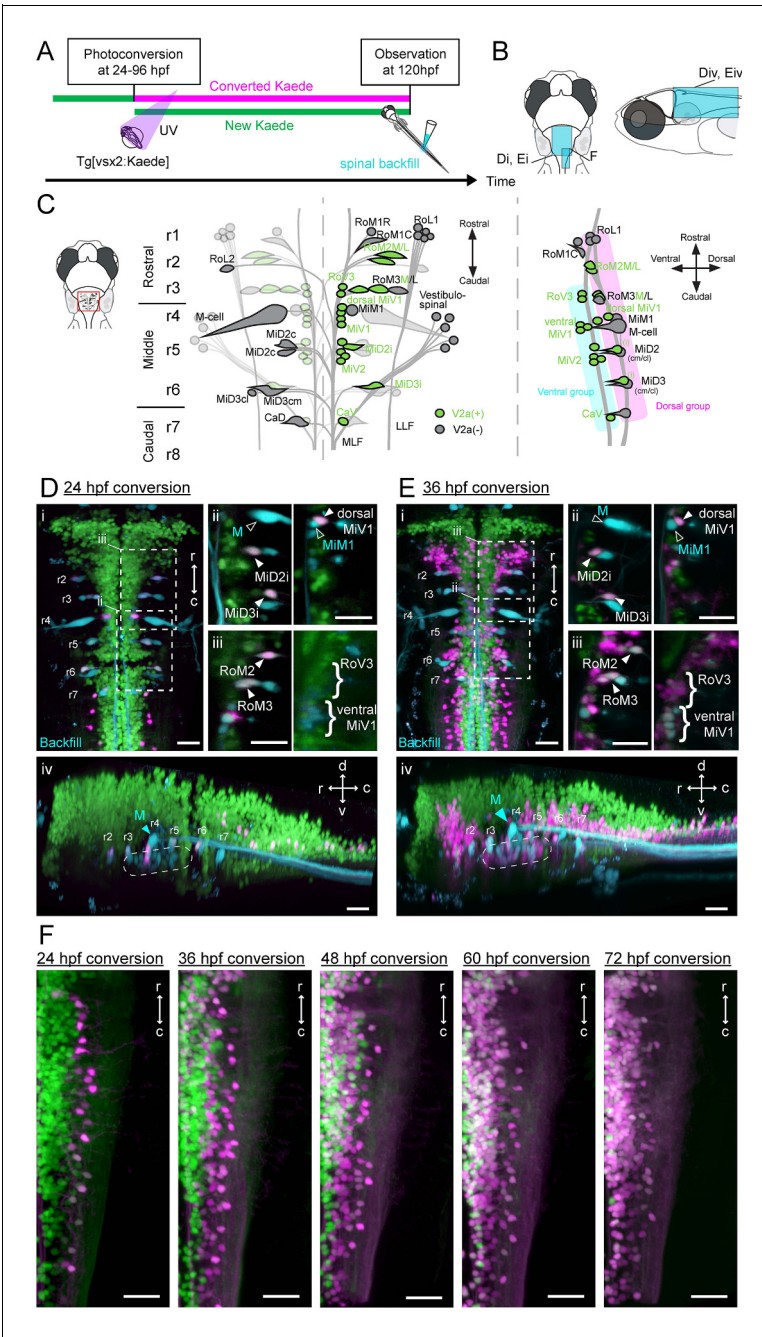

**Figure 2.** Birthdating and identification of hindbrain V2a descending neurons. (**A**) Experimental procedure. (**B**) Regions displayed (cyan patch). (**C**) A schema showing reticulospinal (RS) neurons and its V2a subpopulation in dorsal (left) and lateral (right) views based on *Mendelson (1986)* and *Kimura et al. (2013)*. V2a neurons are indicated in green. Rostral, middle and caudal populations are in rhombomere 1–3 (r1, r2 and r3), 4–6 (r4, r5 and r6) and 7–8 (r7 and r8) respectively. Dorsal and ventral RS group are indicated in purple and cyan rectangles, respectively. (**D**) Hindbrain V2a neurons photoconverted at 24 hpf with spinal backfill of reticulospinal neurons. Green, unconverted Kaede; magenta, photoconverted Kaede; cyan, backfill; M, Mauthner. (**i**) Dorsal view. Dotted rectangles indicate the locations of panels ii and iii. (**ii**) Optical slices showing caudal reticulospinal neurons. (**iii**) Optical slices showing rostral reticulospinal neurons (left panel, dorsal slice; right panel, ventral slice). Filled arrowheads and curly brackets indicate V2a reticulospinal neurons. Open arrowheads indicate non-V2a reticulospinal neurons. (**iv**) Side-view. Dotted rounded rectangle indicates the location of ventral reticulospinal neurons. (**E**) Hindbrain V2a neurons photoconverted at 36 hpf, and reticulospinal neurons labeled with spinal

*Figure 2 continued on next page*

that were labeled by dye injection in the spinal cord (*Figure 2C–E*, n = 4 for each timepoint). These neurons are identifiable across animals and are named based on their rostrocaudal (*Figure 2C*, left) and dorsoventral positions (*Figure 2C*, right) (*Kimmel et al., 1982*; *Mendelson, 1986*). The dorsal subpopulation of reticulospinal neurons (*Figure 2C*, MiD2i, MiD3i, RoM2, RoM3 and dorsal MiV1) corresponded to V2a neurons born by 24 hpf (*Figure 2D*), while the majority of ventral reticulospinal neurons (RoV3, ventral MiV1 and MiV2) corresponded to V2a neurons born by 36 hpf (*Figure 2E*; *Video 2*). This birth order is consistent with previous birthdating analysis of reticulospinal neurons based on the incorporation of a marker into replicating DNA (*Mendelson, 1986*). Other than the large spinal projection neurons that are labeled consistently by dye injection, a majority of the V2a neurons in the caudal hindbrain have also been shown to project to the spinal cord (*Kimura et al., 2013*) (*Video 1*). In this region, the cell bodies of the earliest-born V2a neurons were located lateral to the populations born afterwards (*Figure 2F*). This clear topographical organization allows us to readily identify the earliest-born population and the later-born population in the caudal hindbrain. Notably, this region contained the youngest group of neurons among the hindbrain V2a descending neurons we examined, suggesting their roles in the refined locomotion that develops later (*Figures 2F* and 48 hpf conversion).

For the hindbrain V2a neurons to contribute to the development of the locomotor repertoire, they first need to establish connections to spinal neurons. To examine the development of spinal projections we optically back labeled V2a descending neurons from the spinal cord ('optical backfill') using photoconversion restricted to the rostral spinal cord (*Kimura et al., 2013*). This procedure was repeated at multiple developmental time points to examine the developmental sequence of spinal projections (*Figure 3A*, n = 6 for each timepoint). The optical backfill at 36 hpf labeled the dorsal V2a but not the ventral V2a reticulospinal neurons (*Figure 3C*) while the backfill at 60 hpf labeled both the dorsal and ventral reticulospinal neurons (*Figure 3D*), indicating that these neurons had projected to the rostral spinal cord in sequence based on their respective differentiation times. The V2a subpopulation in the caudal hindbrain showed a pattern similar to that indicated by the birthdating analysis (*Figure 3E*). This indicates that the sequential development of spinal projections based on differentiation time holds true for the caudal hindbrain V2a subpopulation as well. Among these descending neurons, the caudal medial V2a neurons were the last to project to the spinal cord. As these neurons had descending processes in the rostral spinal cord by 84 hpf, it is plausible that they contribute to the refined locomotion that appears as early as 96 hpf.

Collectively, the results of our developmental analysis showed the timeline of neurogenesis and spinal projections of hindbrain V2a descending neurons relative to the development of the locomotor repertoire. Furthermore, we found that the birth order of these neurons dictated the positions of their cell bodies and the order of their spinal projections. Most importantly, this analysis established a way to identify the birthdates of these neurons based on the positions of their cell bodies. This set the stage for functional analysis of these neurons, wherein we characterized how neurons of different age were recruited during the distinct locomotor patterns that fish develop in sequence so as to examine the functional links between the development of V2a neurons and the development of locomotor behaviors.

## Hindbrain V2a neurons of different ages are recruited differentially during distinct locomotor patterns

To identify functional cell groups related to the crude and refined locomotor behaviors that appear in sequence during development, we examined the recruitment of hindbrain V2a neurons expressing the calcium indicator GCaMP6s (*Chen et al., 2013*) using whole-hindbrain two-photon volumetric imaging at 2 Hz in 120–132 hpf fish, which are capable of generating both forms of locomotion. Fish were paralyzed and axial motor activity was monitored using glass pipettes attached to motor nerves innervating axial muscles on both sides (*Figure 4*; *Video 3*). In this condition, fish spontaneously exhibited episodes of rhythmic axial motor activity (*Figure 4A* i; *Figure 4B* i). The maximum beat frequencies of these episodes matched those of slow spontaneous swims generated through tail-restricted movements in unparalyzed fish (12 fish, *Figure 4C*) (*Mirat et al., 2013*; *Marques et al., 2018*), suggesting that these episodes are fictive analogs of refined locomotor behavior. To elicit the crude locomotor behaviors, we used two stimulus paradigms (six fish for each paradigm). The first paradigm made use of a transient electrical pulse delivered to one side of the head to evoke locomotor episodes similar to escapes in unparalyzed condition (*Figure 4A* i). These

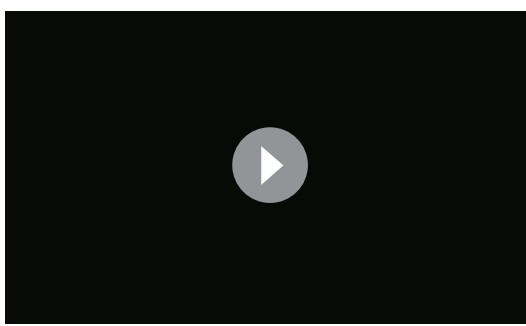

**Video 2.** Hindbrain V2a neurons born before and after 36 hpf. A series of horizontal slices showing hindbrain V2a neurons born before 36 hpf (magenta) and those born after 36 hpf (green) based on the photoconversion of Kaede expressed under vsx2 promoter. The confocal stack was acquired at five dpf and then registered to ZBB atlas. Key cell groups are highlighted as the horizontal slice moves from dorsal to ventral. Magenta, photoconverted Kaede; green, unconverted kaede; gray, brain structure volume. Rostral to the left.

DOI: https://doi.org/10.7554/eLife.42135.007

episodes started with strong axial motor activity on the side contralateral to stimulation, which is consistent with the powerful whole-body bending away from the stimulus observed at the start of escape (*Liu and Fetcho, 1999*; *Koyama et al., 2016*). Maximum beat frequencies during these episodes was also significantly higher than those during spontaneous rhythmic activity (*Figure 4C*, p<0.001). This is consistent with fast rhythmic bending observed during escape (*Mirat et al., 2013*); *Marques et al., 2018*), further suggesting that these episodes correspond to escapes. The second paradigm employed a gradual mechanical stimulus applied to the front of the head to evoke locomotor episodes similar to struggles (*Liao and Fetcho, 2008*). These episodes exhibited powerful bursting activity (*Figure 4B* i), consistent with the powerful whole-body bending observed during struggles (*Liao and Fetcho, 2008*). The maximum beat frequencies of these powerful bouts of bursting activity were clearly lower than those of putative escapes (*Figure 4C*, p<0.001) and also statistically lower than those of spontaneous slow swims (*Figure 4C*, p<0.001). They also matched the frequencies previously reported for struggles

(*Liao and Fetcho, 2008*). Peripheral motor nerve recordings from a single site on each side did not allow us to determine if the rostrocaudal patterns of axial motor activity were organized similarly to those observed in unparalyzed fish, but beat frequency and relative strength of axial motor activity were similar to those of axial movements observed in unparalyzed fish. With these three distinct locomotor patterns, we examined not only how hindbrain V2a descending neurons are recruited during crude forms of locomotion that develop early and more refined form of locomotion that develop later but also how the recruitment of these neurons changes in relationship to beat frequency and strength of axial motor activity.

To examine neuronal recruitment, we first analyzed imaging data, focusing on the V2a descending neurons we had identified based on development. In the electrical shock paradigm (*Figure 4A*), almost all the identifiable dorsal neurons were recruited during putative escapes but primarily on the side ipsilateral to the leading axial motor activity (*Figure 4A* iii, blue traces). This is consistent with the primarily ipsilateral projections of these neurons (*Kinkhabwala et al., 2011*; *Kimmel et al., 1982*). On the other hand, the ventral MiV1 and caudal medial neurons showed stronger activity during weak spontaneous swimming. In the gradual mechanical stimulus paradigm, a subset of the neurons recruited during putative escapes were recruited when there was strong 'struggle' like motor activity on the ipsilateral side (*Figure 4B* iii-iv, blue traces close to the end). The ventral MiV1 and caudal medial cells were again recruited during weak spontaneous swimming. When we sorted these cell types based on their birthdate and examined their responses during these motor patterns, it became apparent that there was a systematic relationship between recruitment and birthdate (*Figures 4C* and *6* fish for each paradigm). The early-born group (*Figure 4C*,<24 hpf) did not show any clear response during spontaneous weak swimming, but many neurons in this group showed significant responses during putative struggles that exhibited strong locomotor activity but with low beat frequency. The responses in a few of these neurons were even stronger during putative escapes that exhibited strong locomotor activity with high beat frequency (*Figure 4C*, MiD2i, RoM2 and RoM3). The recruitment of the early-born group during both putative struggles and escapes indicates that this group of neurons underlies powerful whole-body movements observed in both struggles and escapes in the unparalyzed condition. The further recruitment of this group during putative escapes also indicates that such enhanced activity may lead to fast cyclic movements of the whole-body as observed in escapes. On the other hand, the intermediate-aged group (*Figure 4C* 24-36 hpf) showed mixed recruitment patterns; the subgroup in the rostral hindbrain (*Figure 4C*, RoV3)

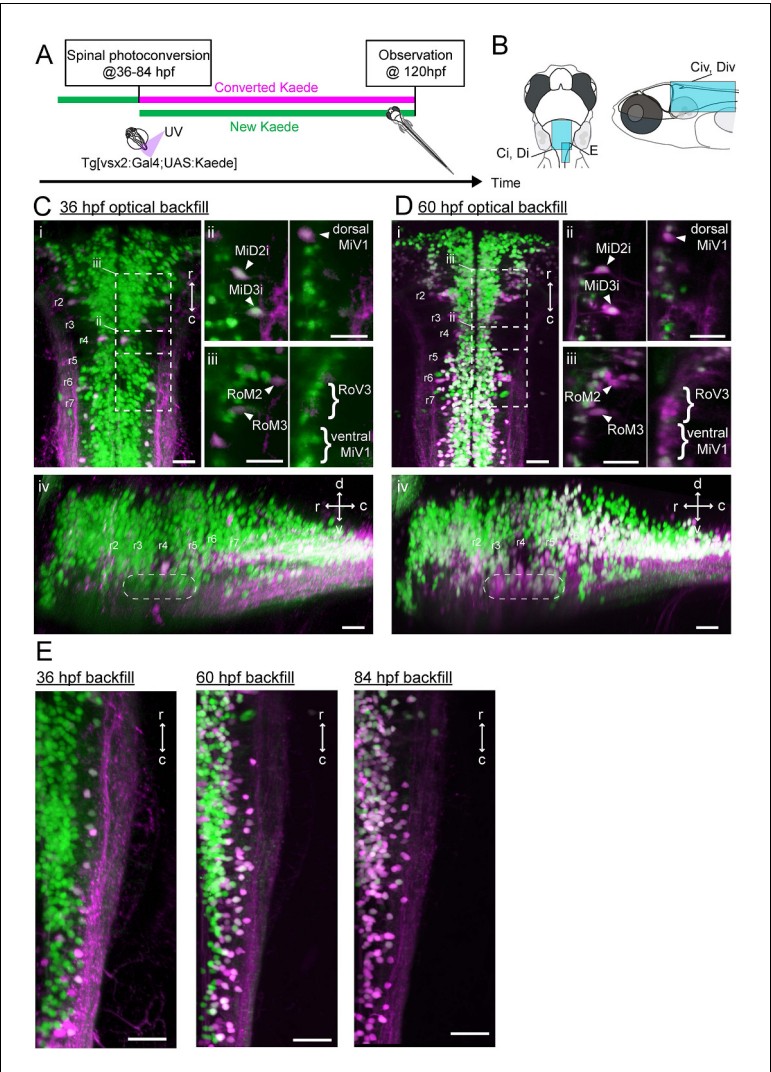

**Figure 3.** Development of spinal projections of hindbrain V2a descending neurons. (**A**) Experimental procedure. (**B**) Regions displayed. (**C**) Hindbrain V2a neurons optically backfilled from the rostral spinal cord at 36 hpf. Magenta, photoconverted Kaede; green, unconverted Kaede. (**i**) Dorsal view. Dotted rectangles indicate the locations of panels ii and iii. (**ii**) Optical slices showing caudal V2a reticulospinal neurons. (**iii**) Optical slices showing rostral V2a reticulospinal neurons (left panel, dorsal slice; right panel, ventral slice). (**iv**) Side-view. Dotted rounded rectangle indicates the position of ventral reticulospinal neurons. Hindbrain segments (r2 to r7) were identified based on V2a reticulospinal neurons. (**D**) Hindbrain V2a neurons optically backfilled from rostral spinal cord at 60 hpf. Photoconverted Kaede is shown in magenta and unconverted Kaede in green. (**i**) Dorsal view. Dotted rectangles indicate the locations of panels ii and iii. (**ii**) Optical slices showing caudal reticulospinal neurons. (**iii**) Optical slices showing rostral reticulospinal neurons (left panel, dorsal slice; right panel, ventral slice). (**iv**) Side-view. Dotted rounded rectangle indicates the location of ventral reticulospinal neurons. (**E**) Dorsal views of caudal hindbrain V2a neurons optically backfilled at 36, 60 and 84 hpf. r, rostral; c, caudal; d, dorsal; v, ventral; Scale bars, 30 μm.

DOI: https://doi.org/10.7554/eLife.42135.008

showed comparably strong activity during putative escapes and putative struggles but not during weak spontaneous swims whereas the subgroup in the middle hindbrain (*Figure 4C*, ventral MiV1) showed comparably strong activity in all three conditions. In contrast, the late-born group in the caudal hindbrain (*Figure 4C* 36-60 hpf) showed more activity only during weak spontaneous swimming. Interestingly, during stimulus-evoked strong swims, these neurons showed decreased activity instead, raising the possibility of inhibition during strong swims. Taken together, our findings

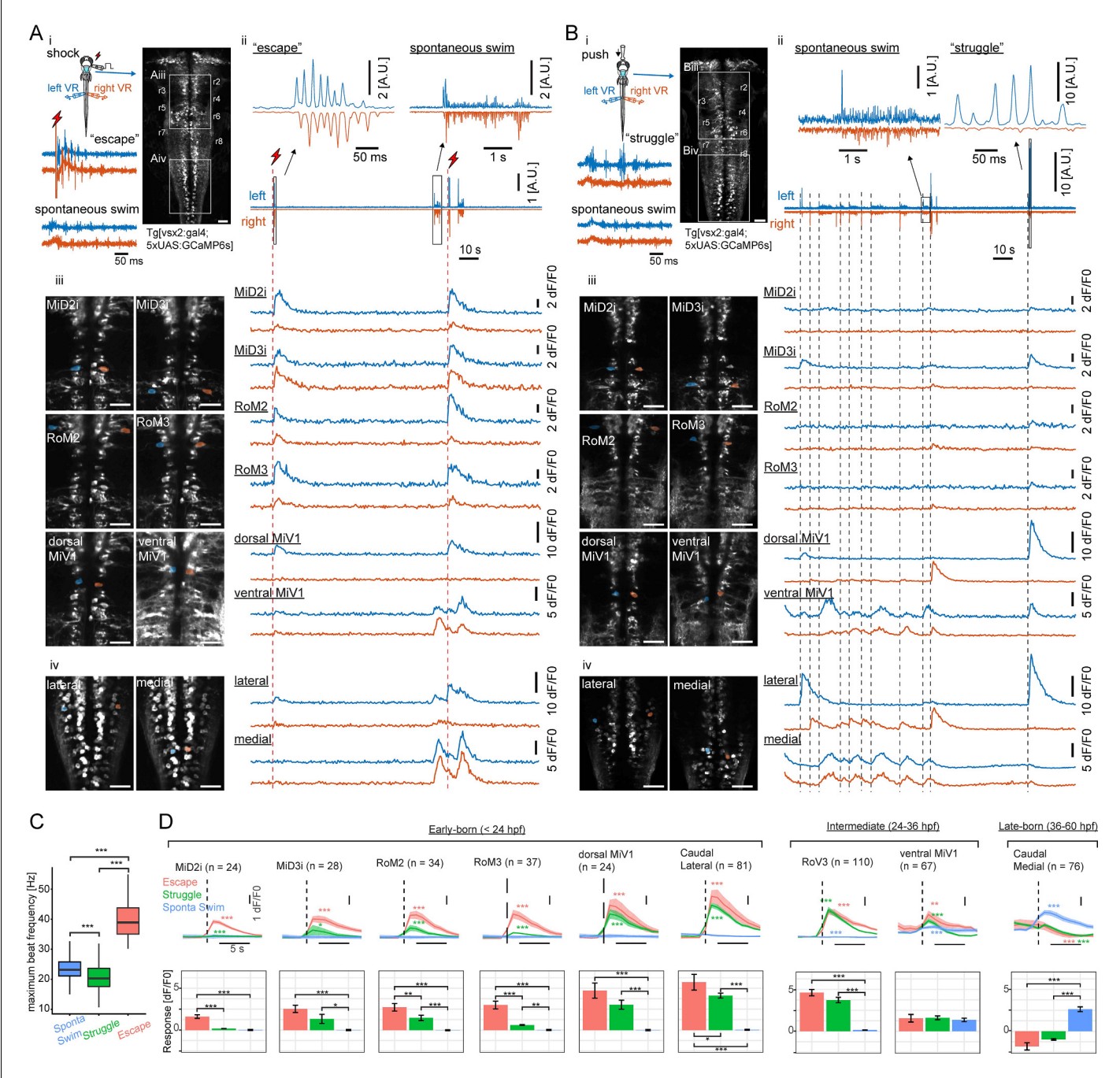

**Figure 4.** Recruitment of hindbrain V2a neurons during distinct locomotor behaviors. (**A**) Ca²⁺ responses of hindbrain V2a neurons during putative escape induced by a transient electrical stimulus to the head. (**i**) Experimental setup. The image on the right shows the dorsal view of hindbrain V2a neurons expressing GCaMP6s. White squares indicate the regions displayed in subsequent panels. Hindbrain segments (r2 to r8) were identified based on identifiable V2a reticulospinal neurons. Example raw traces from left and right ventral root (VR) activity during a putative escape and spontaneous swim episode (blue trace, left VR signal; orange trace, right VR signal; red thunder symbol, onset of an electrical pulse). (**ii**) Example traces of processed swim signals during a spontaneous weak swim episode and shock-induced strong swim episode. (blue trace, processed left VR; orange trace, processed right VR; red thunder mark, shock onset). (**ii-iii**) Location of hindbrain V2a neurons and corresponding Ca²⁺ traces during the locomotion patterns shown in panel i (blue, left ROIs and their Ca²⁺ responses; orange, right ROIs and their Ca²⁺ responses). Vertical dotted red lines indicate the onsets of the shock stimuli. (**ii**) Ca²⁺ traces of rostral V2a neurons (**iii**) Ca²⁺ traces of caudal hindbrain V2a neurons. (Scale bars in the images, 30 μm). (**B**) Ca²⁺ responses of hindbrain V2a neurons during a putative struggle induced by the gradual mechanical stimulus. The panels are organized as in A. Note the strong and slowly alternating (between left and right) axial motor activity in panel i. (**C**) Distribution of maximum tail beat frequency in each locomotor pattern (Spontaneous swims, 10,691 episodes; Struggle, 214 episodes; Escape, 183 episodes). Significant differences in maximum tail beat

*Figure 4 continued on next page*

*Figure 4 continued*

frequency across three locomotor patterns are indicated with asterisks (Dunn's test, \*\*\*p<0.001). (**D**) Locomotion event-triggered average $Ca^{2+}$ responses of hindbrain V2a neurons sorted based on their birthdate. Top, Time course of the event-triggered average $Ca^{2+}$ response (Orange trace, putative escape; Green trace, putative struggle; Blue trace, spontaneous slow swim). Shaded area indicates standard error across replicates for each cell type (n, number of cells). Significant changes from the baselines are indicated with asterisks (paired t-test, \*p<0.05, \*\*p<0.01, \*\*\*p<0.001). Bottom, Peak amplitude of locomotion event-triggered average $Ca^{2+}$ response (Orange bar, putative escape; Green bar, putative struggle; Blue bar, spontaneous slow swim). Significant differences in response amplitude across conditions are indicated with asterisks (Tukey test, \*p<0.05, \*\*p<0.01, \*\*\*p<0.001).

DOI: https://doi.org/10.7554/eLife.42135.009

The following source data is available for figure 4:

**Source data 1.** Contains numerical data plotted in *Figure 4D*.

DOI: https://doi.org/10.7554/eLife.42135.010

indicate that a recruitment pattern that is based on the time of differentiation holds true for all the hindbrain V2a descending neurons that we identified based on development; the early-born group is recruited during the crude forms of locomotion (escapes and struggles) that appear early in development whereas the late-born group is recruited during the refined locomotion that appears later in development. This finding, along with the finding that spinal projections develop sequentially, suggests that distinct functional groups emerge in developmental sequence and contribute to the development of locomotor behaviors. Furthermore, the active set of hindbrain V2a neurons switched based on the type of ongoing motor activity, which is reminiscent of the recruitment of spinal interneurons (*McLean et al., 2008*). This raises the possibility that distinct functional groups of hindbrain V2a neurons may provide excitatory drive to corresponding spinal functional groups.

To see if our observations about neuronal recruitment extend to the complete set of hindbrain V2a neurons, we performed regression analysis (*Miri et al., 2011b*) (*Figure 5*). In the electrical pulse paradigm, we used two regressors to distinguish the activity related to shock-induced putative escapes from activity related to weak spontaneous swimming (*Figure 5A* ii). The activity map for the shock-evoked putative escapes (*Figure 5A* iii, left; *Figure 5A* iv, magenta; *Video 4*, magenta) revealed a large population of recruited neurons on the side contralateral to the stimulus (ipsilateral to the leading side of motor activity), consistent with their primarily ipsilateral projections (*Kinkhabwala et al., 2011*). This includes the dorsal early-born group and the rostral intermediate group (RoV3) in the rostral hindbrain (*Figure 5A* v, magenta) and a large number of the lateral early-born neurons in the caudal hindbrain (*Figure 5A* vi, magenta). The activity map for spontaneous weak swimming

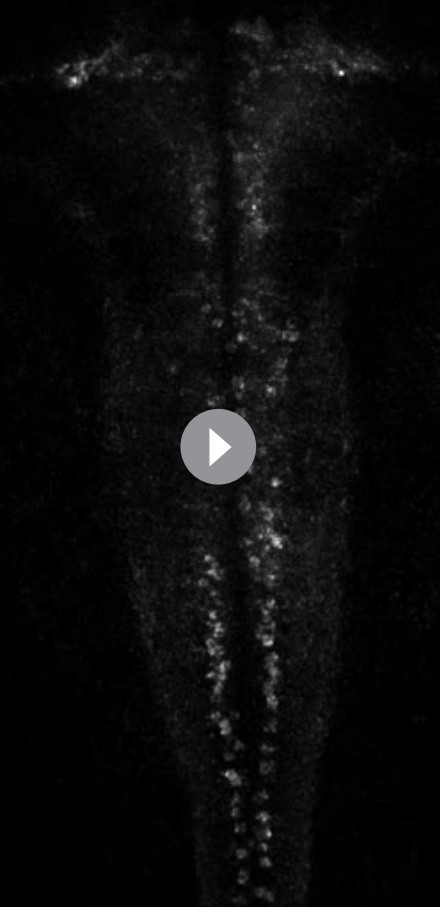

**Video 3.** $Ca^{2+}$ activity of hindbrain V2a neurons during fast and slow swimming episodes. Dorsal view of GCaMP6s-labeled hindbrain V2a neurons showing activity correlated with distinct locomotor patterns. Dorsal side of the brain is at the top of movie frames. Axial motor activity is encoded with a sound signal. The time of the electrical stimulus delivered to the right side is indicated by a red square patch in the lower right side. The movie is sped up 7.5 times. Rostral to the top.

DOI: https://doi.org/10.7554/eLife.42135.011

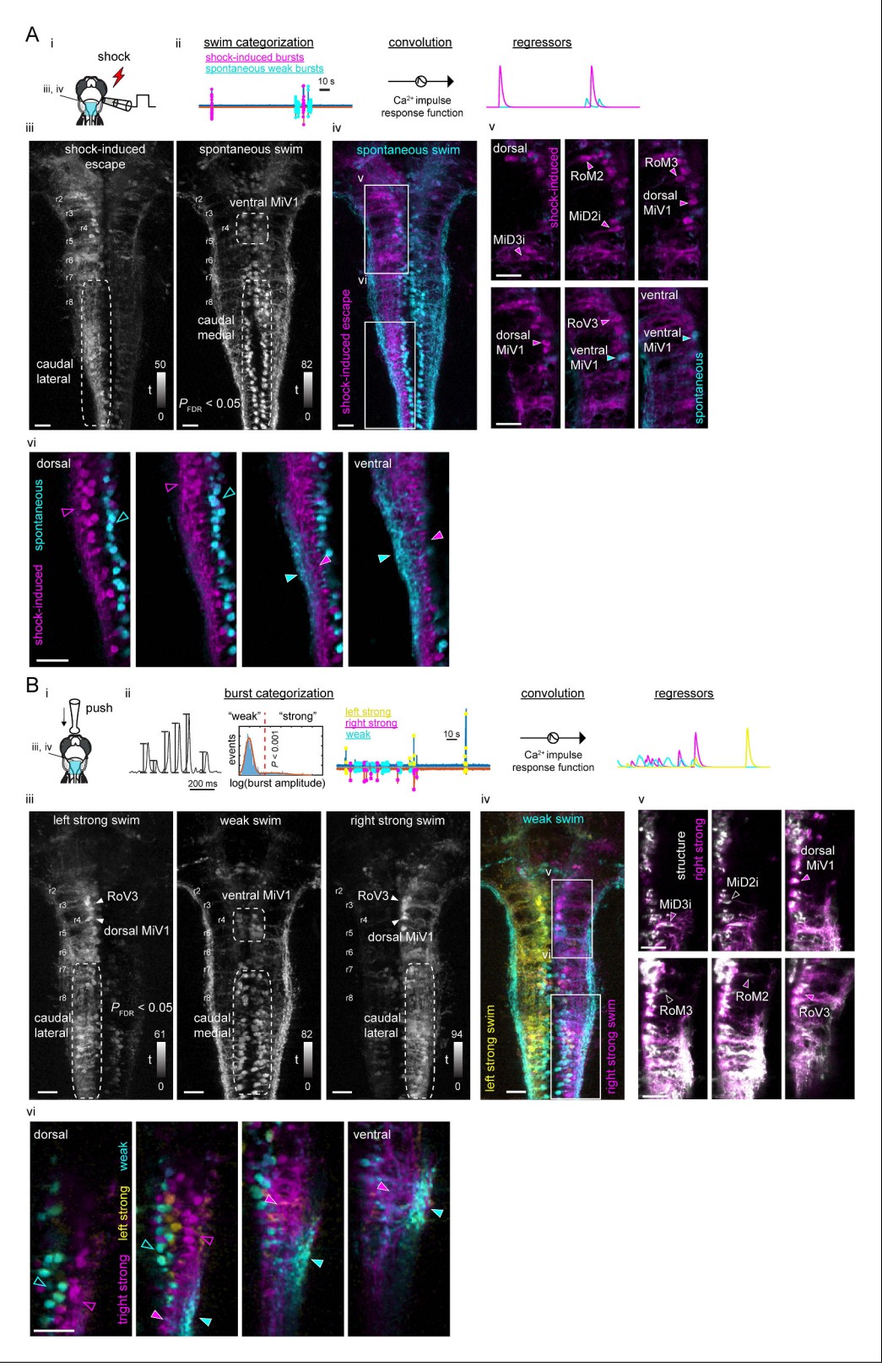

**Figure 5.** Functional segregation of the neuropil of hindbrain V2a neurons as revealed by regression analysis. (**A**) Regression analysis of hindbrain V2a neurons in the transient electrical stimulus (shock) experiment. (**i**) Experimental setup and the region displayed (cyan patch). (**ii**) Regressors used to create activity maps (see Materials and methods). (**iii**) Dorsal view of activity maps for each regressor. T-value was coded in grayscale and thresholded at $P_{FDR\ (False\ Discovery\ Rate)} < 0.05$ (see Materials and methods). Dotted round rectangles indicate distinct caudal hindbrain populations revealed by each

*Figure 5 continued on next page*

*Figure 5 continued*

regressor. (**iv**) Overlay of activity maps (cyan, spontaneous swim; magenta, shock-induced fast swim). Rectangles indicate the locations of the images in panel v and vi. (**v**) Optical slices of the overlaid activity maps in the rostral hindbrain (magenta, shock-induced swim; cyan, spontaneous swim). Arrowheads indicate reticulospinal neurons identified in each map. (**vi**) Optical slices of the overlaid activity maps in the caudal hindbrain (magenta, shock-induced swim; cyan, spontaneous swim). Arrowheads highlight the segregation of spontaneous swim related signal (cyan open arrowhead, cell bodies; cyan filled arrowhead, neuropil) and shock-induced swim related signal (magenta open arrowhead, cell bodies; magenta filled arrowhead, neuropil). Scale bars, 30 µm. (**B**) Regression analysis of hindbrain V2a neurons in the gradual mechanical stimulus (push) experiment. (**i**) Experimental setup and the region displayed (cyan patch). (**ii**) Regressors used to create activity maps (see Materials and methods). (**iii**) Dorsal view of activity maps for each regressor. T-value was coded in grayscale and thresholded at $P_{FDR}$ <0.05. Dotted round rectangles and arrowheads are used to highlight structures revealed in each activity map. (**iv**) Overlay of activity maps (cyan, weak swim; yellow, left strong swim; magenta, right strong swim). Rectangles indicate the locations of the images in panel v and vi. (**v**) Optical slices of the activity map for the right strong swim (magenta) overlaid on the structural image of V2a neurons (white) in the rostral hindbrain (magenta arrowheads, recruited reticulospinal neurons; white open arrowheads, non-recruited reticulospinal neurons). (**vi**) Optical slices of the overlaid activity maps in the caudal hindbrain (magenta, right strong swim; cyan, weak swim). Arrowheads highlight the segregation of weak swim-related signal (cyan open arrowhead, cell bodies; cyan filled arrowhead, neuropil) and strong swim-related signal (magenta open arrowhead, cell bodies; magenta filled arrowhead, neuropil). Scale bars, 30 µm.

DOI: https://doi.org/10.7554/eLife.42135.012

The following figure supplement is available for figure 5:

**Figure supplement 1.** Activity maps from multiple subjects in the rostral and the caudal hindbrain.

DOI: https://doi.org/10.7554/eLife.42135.013

revealed the middle intermediate group (ventral MiV1) in the rostral hindbrain (*Figure 5A* iii, right; *Video 4*, green) and a large number of the medial late-born neurons in the caudal hindbrain (*Figure 5A* iii, right; *Figure 5A* vi, cyan; *Video 4*, green). For the gradual mechanical stimulus paradigm that produced putative struggles that varied in latency, we constructed three regressors based on the strength of the axial motor activity: two regressors for the strong motor activity for each side to capture the struggle related activity and one regressor for the weak motor activity to capture the activity related to spontaneous weak swims (*Figure 5B* ii). The activity maps for the strong motor activity (*Figure 5B* iii, left and right; *Figure 5B* iv, yellow and magenta; *Video 5*, yellow and magenta) revealed a large population of neurons ipsilateral to the side of motor activity (*Figure 5B* iii, left and right) consisting of a subpopulation of the dorsal early-born group, the rostral intermediate group (RoV3) in the rostral hindbrain (*Figure 5B* v), and a large number of the early-born lateral neurons in the caudal hindbrain (*Figure 5B* vi, magenta). The map for spontaneous swimming revealed the same set of the neurons shown for spontaneous swimming occurring in the electrical pulse paradigm (*Figure 5B* iii, middle; *Figure 5B* iv, cyan; *Video 5*, cyan). These regression maps were reproducible across fish (*Figure 5—figure supplement 1*) and further support the differentiation time dependent recruitment pattern revealed by the ROI analysis. The activity maps for putative escapes and struggles were surprisingly similar, especially in the caudal hindbrain (*Figure 5—figure supplement 1*). This supports the idea that the early-born population contributes to the crude whole-body movements observed in both struggles and escapes. However, this analysis also made it clear that the activity of the early-born group is further enhanced during escapes compared to struggles, especially in the rostral hindbrain (*Figure 5—figure supplement 1*), suggesting again that further activation of this population leads to fast cyclic movements of the whole-body, as observed in escapes. Interestingly, these maps also revealed a striking functional separation of neuropil in the caudal hindbrain (*Figure 5A* vi; *Figure 5B* vi; *Figure 5—figure supplement 1C*). The neuropil active during weak spontaneous swimming was located lateral to the neuropil active during putative struggles (*Figure 5B* vi, filled arrowheads; *Figure 5—figure supplement 1C*, Struggle, arrowheads). This functional segregation is the opposite of the functional segregation of cell bodies in the caudal hindbrain (*Figure 5B* vi, open arrowheads). This neuropil separation was maintained when additional cells were recruited for shock-induced escapes (*Figure 5A* vi, filled arrowheads; *Figure 5—figure supplement 1C*, Escape, arrowheads), suggesting that these additional cells also followed this functional separation. Over all, this raises the possibility that these functional neuronal groups project their descending axons through distinct regions in the neuropil.

To this point, we showed a correspondence between the birth order of hindbrain V2a neurons and their involvement in distinct forms of locomotion that appear in sequence. This suggests that distinct functional cell groups emerge in the hindbrain and project to the spinal cord in sequence

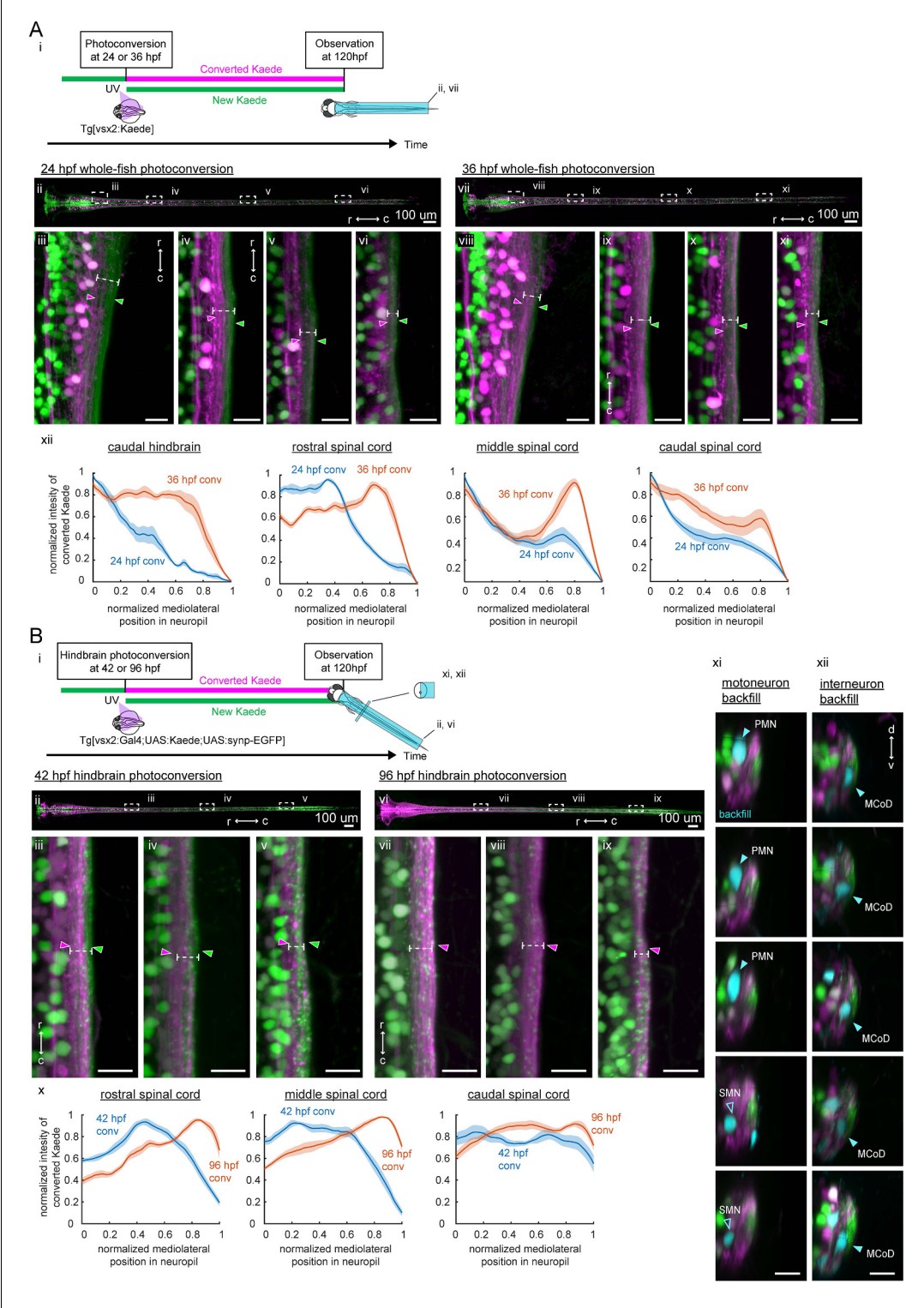

**Figure 6.** Birthdate-related segregation of spinal projections from hindbrain V2a neurons. (**A**) Birthdate-related segregation of the neuropil of V2a neurons in the hindbrain and the spinal cord. (**i**) Experimental procedure and region displayed (gray patch). Converted Kaede is shown in magenta and unconverted Kaede is shown in green. (**ii-vi**) Segregation of neuropil from V2a neurons born before and after 24 hpf. (**ii**) Dorsal view of the hindbrain and the spinal cord. Dotted rectangles indicate the locations of the images in the following panels. (**iii-vi**) Close-up views of the early-born neuropil (magenta arrowheads) and the late-born neuropil (green arrowheads) in the hindbrain and spinal cord. Dotted lines indicate the mediolateral extent of neuropil. (**iii**) Dorsal view of the caudal hindbrain. (**iv**) Dorsal view of the rostral spinal cord. (**v**) Dorsal view of the middle spinal cord. (**vi**) Dorsal view of

*Figure 6 continued on next page*

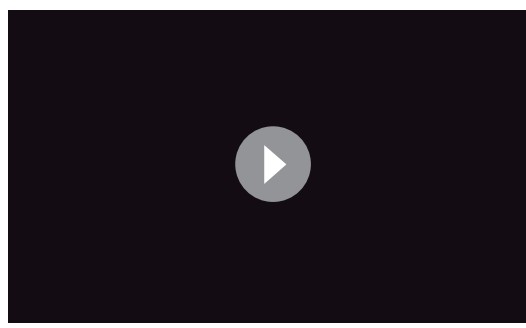

**Video 4.** Activation maps in the shock paradigm. Activation maps for shock-induced escape (magenta) and spontaneous swim (green) registered to ZBB atlas overlaid on a brain structure volume (gray). Key cell groups are highlighted as the horizontal slice moves from dorsal to ventral. Rostral to the left.
DOI: https://doi.org/10.7554/eLife.42135.014

and contribute to the diversification of locomotor repertoire. Furthermore, the similarity in the recruitment of hindbrain and spinal functional groups as well as the spatial segregation of the neuropil of the hindbrain functional groups both raise the possibility that parallel pathways connect matching sets of hindbrain and spinal functional groups. Having established a link between the birth order and function of hindbrain V2a neurons, we next examined the spinal organization of descending processes from hindbrain functional groups.

## Chronological layering of spinal projections of hindbrain V2a neurons suggests parallel descending pathways organized by birthdate

To understand how the spinal projections from hindbrain V2a neurons are organized to support the development of the locomotor repertoire, we examined if the aforementioned sequentially generated functional groups of hindbrain V2a neurons exhibit distinct innervation patterns to the spinal cord. First, we examined the overall neuropil organization of hindbrain and spinal V2a neurons based on the time of differentiation by photoconverting early-born V2a neurons at 24 and 36 hpf (*Figure 6A*, n = 6 fish for each timepoint). In the hindbrain (*Figure 6A* iii), and throughout the spinal cord (*Figure 6A* iv-vi), the neuropil from V2a neurons born before 24 hpf (*Figure 6A* iii-vi, magenta arrowheads) was located medial to the neuropil from neurons born after 24 hpf (*Figure 6A* iii-vi, green arrowheads), consistent with the functional segregation of neuropil we observed in the caudal hindbrain. The neuropil from V2a neurons born before 36 hpf was still located medial to the neuropil from neurons born after 36 hpf (*Figure 6A* viii-xi, arrowheads). However, it extended more toward the lateral surface than the neuropil from V2a neurons born before 24 hpf (*Figure 6A* xii). This suggested that axonal processes from new neurons were continuously being added laterally to preexisting processes, forming layers based on the time of differentiation.

To see if this neuropil organization was maintained in the descending processes of hindbrain V2a neurons, we photoconverted V2a neurons only in the hindbrain (*Figure 6B*). The Gal4-UAS system was used to gain enough expression of Kaede to be able to visualize the spinal processes of hindbrain V2a neurons. Putative presynaptic terminals were also labeled with synaptophysin-GFP driven by the Gal4-UAS system. To visualize the spinal projections from only the early-born and intermediate-aged hindbrain V2a groups, UV light was shone on the hindbrain at 42 hpf (*Figure 6B* ii-v, n = 8 fish). This conversion time was selected to account for the delay in the expression of Kaede driven by the Gal4-UAS system and to match the age groups labeled by the photoconversion of Tg (vsx2:Kaede) at 36 hpf. The descending processes from the early-born and intermediate-aged groups reached the caudal part of the spinal cord (*Figure 6B* v) and were in the medial part of the neuropil in the rostral and middle spinal cord (*Figure 6B* iii-iv, magenta arrowheads). To visualize the spinal projections from almost all the hindbrain V2a neurons, we photoconverted Kaede in the hindbrain at 96 hpf and imaged

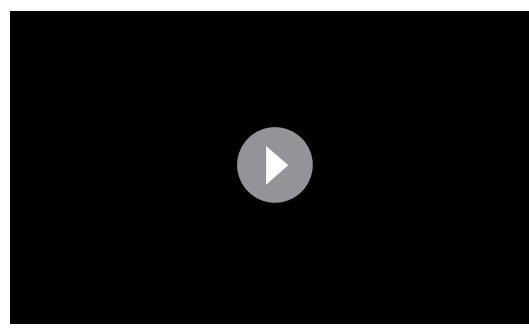

**Video 5.** Activation maps in the push paradigm. Activation maps for push-induced left strong bend (magenta), right strong bend (yellow) and spontaneous swim (cyan) registered to ZBB atlas overlaid on a brain structure volume (gray). Key cell groups are highlighted as the horizontal slice moves from dorsal to ventral. Rostral to the left.
DOI: https://doi.org/10.7554/eLife.42135.015

them at 120 hpf (*Figure 6Bvi-ix*, n = 8 fish). The descending processes from hindbrain V2a neurons covered almost all the neuropil region formed by V2a neurons (*Figure 6B* vii-ix, magenta arrowheads). Comparison of the lateral extents of the descending processes of the early-born and intermediate-aged groups (*Figure 6B* x, blue lines) and all groups (*Figure 6B* x, orange lines) indicated that the descending processes from the late-born group were most prominent in the lateral neuropil in the rostral and middle spinal cord. Taken together, these results indicated that the spinal projections from hindbrain V2a neurons were also layered based on the time of differentiation and that the late-born group had shorter descending processes than the early-born and intermediate-aged groups. The difference in the axon length across different age groups is consistent with the pattern previously revealed by single-cell labeling (*Kinkhabwala et al., 2011*). These observations support the notion that the sequentially generated hindbrain functional groups provide parallel spinal pathways.

To examine the pattern of connections made by each pathway onto possible postsynaptic spinal neurons, we highlighted the descending processes of early-born and intermediate-aged hindbrain V2a neurons by photoconversion in the hindbrain, and additionally backfilled the following two spinal cell types with a far-red dye. The first cell type is primary motoneurons (PMN) which are among the earliest-born neurons in the spinal cord (*Myers et al., 1986*) (*Figure 6B* xi, n = 8 fish). They are only recruited during the strongest of movements in contrast to secondary motoneurons (SMN) which are also recruited during weaker movements (*McLean et al., 2007*; *Menelaou and McLean (2012)*; *Wang and Brehm, 2017*). The second cell type is multipolar commissural neurons (MCoD), which are later-born excitatory interneurons in the rostral spinal cord (*Figure 6B* xii, n = 8 fish). They are only active during slow swimming and have monosynaptic connections to SMNs in the caudal spinal cord (*McLean et al., 2008*; *Fetcho and McLean, 2010*). We found that PMNs were located close to the medial neuropil region occupied by the spinal projections from the early-born hindbrain V2a population, whereas MCoDs were located close to the lateral neuropil region occupied by the spinal projections from the late-born population. Thus, these results support the notion that there are parallel pathways between the hindbrain and the spinal cord that are separable by differentiation time.

The innervation patterns exhibited by these hindbrain V2a age groups are consistent with the type of axial muscle activity observed during the locomotor patterns that these age groups participate in: direct activation of PMNs throughout the spinal cord by the early-born group is expected to lead to the powerful whole-body bends observed during the crude and strong forms of locomotion (escape and struggle), whereas indirect activation of SMNs in the caudal spinal cord through MCoDs should lead to slow bending of the caudal tail observed during the more refined and weaker locomotion. Thus, these observations raise the possibility that each age group contributes to kinematics specific to the locomotor pattern in which this group participates.

## In-depth analyses of V2a reticulospinal neurons

So far, we experimentally investigated the overall organization of the hindbrain V2a population and the results suggest that birthdate-dependent parallel circuit organization underlies the development of locomotor behaviors. To examine this more rigorously and to gain deeper insights about the biophysical properties of V2a neurons and their descending pathways, we examined various subsets of V2a reticulospinal neurons in depth.

### Neuronal excitability and synaptic inputs in V2a pathways vary in accordance with birthdate-related recruitment pattern

To electrophysiologically examine the birthdate-related recruitment pattern, we focused on MiV1 neurons, a small cluster of V2a reticulospinal neurons in rhombomere 4 (*Figure 7A*), which included neurons varying in birthdate and recruitment pattern (*Figures 2*, *4* and *5*).

First, we examined the spatial organization of neurons in this cluster based on the time of differentiation, as indicated by photoconversion-based birthdating from 25 hpf to 37 hpf (*Figure 7B*, n = 6 fish for each timepoint). We found that newly generated V2a neurons were systematically located ventral to preexisting ones (*Figure 7B* iii), in contrast to a previous study reporting that late-born neurons are located dorsal to early-born neurons except in rhombomere 6 (*Kinkhabwala et al., 2011*). To see if this inverted organization was due to migration, we performed

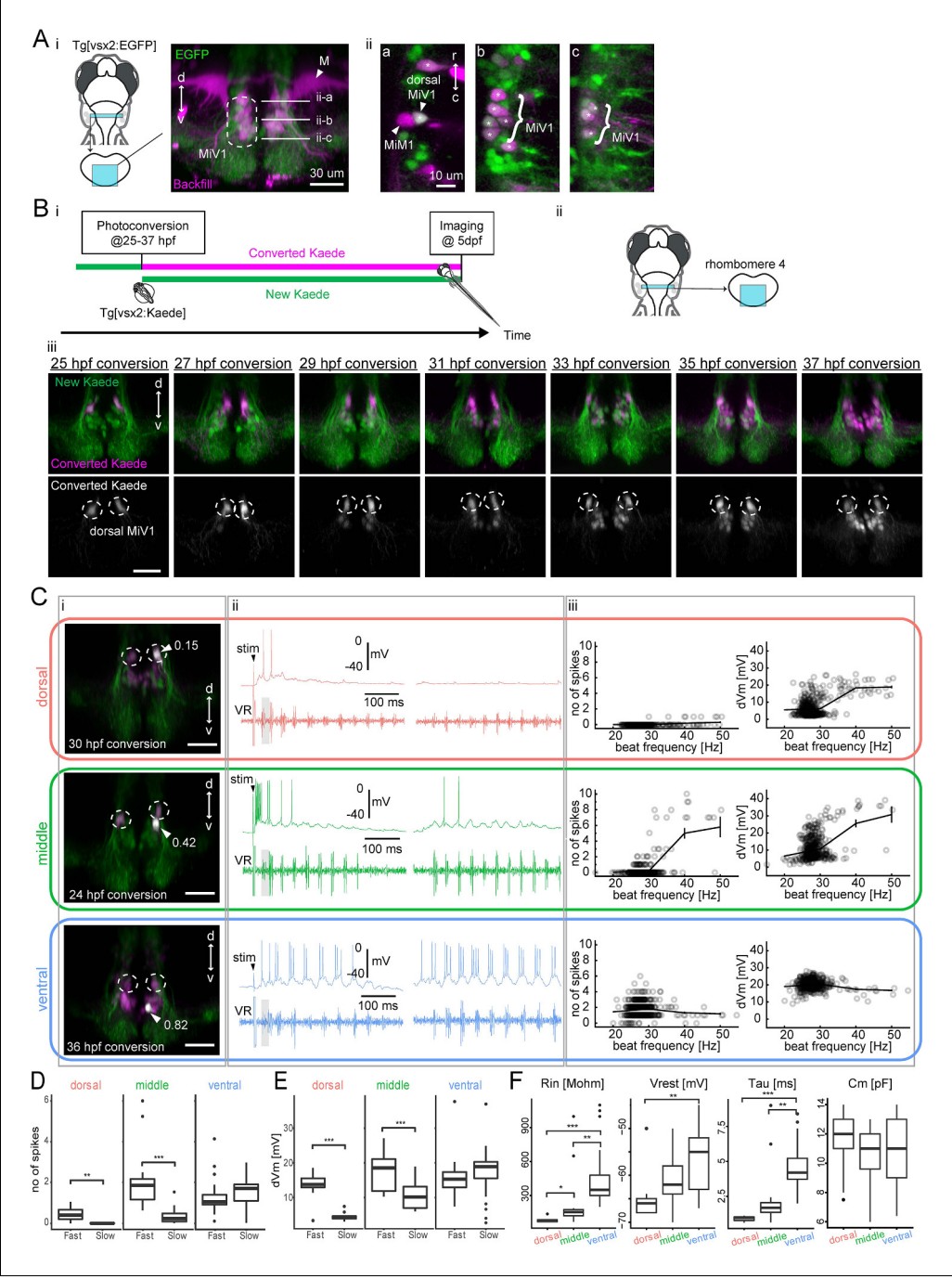

**Figure 7.** Birthdating and electrophysiological analysis of V2a reticulospinal neurons in rhombomere 4. (**A**) V2a reticulospinal (RS) neurons in rhombomere 4. (**i**) Coronal view of V2a reticulospinal neurons in the region of rhombomere four shown by the cyan patch (green, EGFP; magenta, backfill; d, dorsal; v, ventral). Mauthner cell (M) and MiV1 neurons are highlighted. White horizontal lines in the image indicate the optical slices shown in panel ii. (**ii**) Optical slices showing reticulospinal neurons at different depths indicated by white lines in i. Asterisks indicate backfilled V2a neurons. MiM1 and dorsal MiV1 are highlighted with arrowheads. Other more ventral MiV1s are highlighted with curly brackets. (**B**) Birthdate-related topographical organization of V2a RS neurons in rhombomere 4. (**i**) Experimental procedure (**ii**) Region displayed (cyan patch) (**iii**) Coronal views of V2a RS neurons in rhombomere 4 at five dpf showing Kaede photoconverted at a specific time point (25–37 hpf) in magenta in upper panels and gray in lower panels. Dotted circles indicate dorsal MiV1. (**C**) Whole-cell recordings of V2a reticulospinal neurons in rhombomere 4. Top orange row, an early-born dorsal V2a neuron (dorsal); Middle green row, an intermediate V2a neuron (middle); Bottom blue row, a late-born V2a neuron (ventral). (**i**) Coronal view of

*Figure 7 continued on next page*

*Figure 7 continued*

the patched cell (gray, white arrow). Green, unconverted Kaede. Magenta, converted Kaede. Dotted circles indicate dorsal MiV1. The number near the white arrowhead indicates normalized dorso-ventral position of the patched cell (see Materials and methods). (ii) Intracellular activity during shock-induced fast swimming (left) and spontaneous slow swimming (right). An arrowhead indicates the onset of the tail shock. VR, ventral root recording. Gray shaded boxes indicate similar fast beat frequencies. (iii) Number of spikes (left) and membrane depolarization (right) as a function of beat frequency. Open circles are raw data points. The solid line represents mean ± standard error from data binned at 10 Hz intervals. (D) Number of spikes per cycle during fast beat frequency (>35 Hz) and slow beat frequency (<35 Hz) for each age group (dorsal MiV1, n = 8; middle MiV1, n = 13; ventral MiV1, n = 23). Significant differences are indicated with asterisks (**p<0.01, ***p<0.001). (E) Membrane depolarization during fast swim (>35 Hz) and slow beat frequency (<35 Hz) for each age group (dorsal MiV1, n = 8; middle MiV1, n = 13; ventral MiV1, n = 23). Significant modulations based on beat frequency are indicated with asterisks (***p<0.001). (F) Intrinsic properties of each age group (dorsal MiV1, n = 8; middle MiV1, n = 13; ventral MiV1, n = 23). Rin, input resistance; Vrest, resting membrane potential; Tau, membrane time constant; Cm, membrane capacitance. Significant differences are indicated with asterisks (Dunn's test, **p<0.01, ***p<0.001).

DOI: https://doi.org/10.7554/eLife.42135.017

The following source data and figure supplement are available for figure 7:

**Source data 1.** Contains numerical data plotted in *Figure 7D*.
DOI: https://doi.org/10.7554/eLife.42135.019
**Source data 2.** Contains numerical data plotted in *Figure 7E*.
DOI: https://doi.org/10.7554/eLife.42135.020
**Source data 3.** Contains numerical data plotted in *Figure 7F*.
DOI: https://doi.org/10.7554/eLife.42135.021
**Figure supplement 1.** Emergence and migration of hindbrain V2a neurons in rhombomere 4.
DOI: https://doi.org/10.7554/eLife.42135.018

time lapse imaging from 25 hpf to 37 hpf (*Figure 7—figure supplement 1A*, n = 6) and from 48 hpf to 78 hpf (*Figure 7—figure supplement 1B*, n = 6). We found that newly generated neurons were indeed initially located dorsal to preexisting ones (*Figure 7—figure supplement 1A* iii) but the subpopulation of neurons born before 48 hpf migrated ventrally after 48 hpf (*Figure 7—figure supplement 1B* iii, magenta cells; *Video 6*). This supported the idea that the inverted organization we observed in this region was indeed due to radial migration. This analysis revealed a fine-scale birthdate-related topographical organization, which our previous analyses overlooked, and provided an opportunity to examine in detail the relationship between ontogeny and recruitment pattern. For the following recruitment analysis, we used dorso-ventral position as a proxy for birthdate, taking advantage of the fine-scale birthdate-related topographical organization (see Materials and methods). We first identified dorsal MiV1 based on its position just ventral to M-cell, its soma larger than the nearby younger cells, and the presence of photoconverted Kaede. We then defined normalized dorso-ventral position with '0' being the dorsal edge of dorsal MiV1 and '1' being the ventral edge of the most ventral MiV1. Cells other than dorsal MiV1 with the position values lower than 0.5 were categorized as middle MiV1 while those with the position values higher than 0.5 were categorized as ventral MiV1.

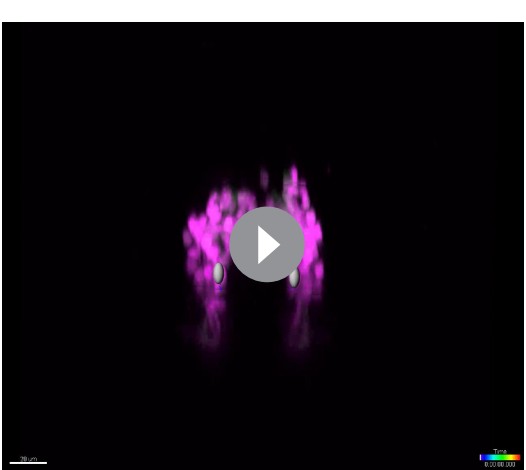

**Video 6.** Migration of hindbrain V2a neurons in rhombomere 4. Coronal view of hindbrain V2a neurons expressing Kaede in rhombomere 4 imaged from 48 hpf to 84 hpf. Kaede photoconverted at 48 hpf is shown in magenta and unconverted Kaede expressed afterwards is shown in green. The earliest-born V2a neurons in rhombomere 4 (dorsal MiV1) are indicated by two gray ellipsoids, one on each side. Note the magenta cells migrating past dorsal MiV1. Dorsal to the top.

DOI: https://doi.org/10.7554/eLife.42135.022

To examine the activity of MiV1 neurons during locomotion, we performed whole-cell recordings in a paralyzed fictive swim preparation (*Figure 7C–F*). We delivered electrical shocks to the tail to induce putative escapes that showed fast frequency bursting activity in axial motor recording (*Figure 7C* ii, left). Locomotor episodes with slower frequency bursting activity were observed spontaneously (*Figure 7C* ii, right) as in the functional imaging experiments. However, whole-cell recordings allowed us to examine how subthreshold and spiking activity changes in relationship to beat frequency within each swim episode. The dorsally located early-born MiV1 neurons (n = 8, position: 0.14–0.18) showed strong depolarization leading to action potentials during the fastest phase of fast swim episodes evoked by electrical shock to the tail (*Figure 7C* ii, left orange traces, shaded cycle). However, they showed no clear depolarization during spontaneous slow swim episodes (*Figure 7C* ii, right orange traces). The middle MiV1 neurons that belong to the intermediate age group (n = 13, position: 0.30–0.48) showed clear spiking activity at the fastest phase of the shock-induced fast swimming. They also showed clear rhythmic depolarization leading to occasional firing during spontaneous slow swimming (*Figure 7C* ii, green traces). The ventral late-born MiV1 neurons (n = 23, position: 0.64–0.93) showed clearly rhythmic firing during both shock-induced fast swimming and spontaneous slow swimming (*Figure 7C* ii, blue traces). Spiking activity showed clear statistical interaction between the time of differentiation and the speed of locomotion (*Figure 7D*, p<0.001), suggesting that groups of neurons that differed by age were recruited differently depending on the speed of locomotion. Indeed, the dorsal and middle MiV1 neurons showed significant modulation of activity based on the speed of locomotion (*Figure 7D*). Subthreshold activity also showed similar interaction (*Figure 7E*, p<0.001), suggesting that this recruitment pattern can be explained at least partly by their distinct synaptic inputs. At the same time, the MiV1 group showed systematic differences in their intrinsic excitability based on their time of differentiation: the older neurons showed lower input resistances and more hyperpolarized membrane potentials compared to younger ones (*Figure 7F*, Rin, Vrest). This is similar to the previously reported relationship between input resistance and differentiation time in the spinal cord and the caudal hindbrain V2a population (*McLean et al., 2007*; *Kinkhabwala et al., 2011*). Furthermore, we also found a systematic change in the membrane time constants: the older neurons showed faster membrane time constants than the younger ones (*Figure 7F*, Tau). This comports well with the selective recruitment of the older neurons during fast tail beats. Thus, taking advantage of whole-cell recordings, we showed that these V2a reticulospinal neurons exhibited a systematic change in the spiking and subthreshold activity within each swim episode, depending on the speed of locomotion. Furthermore, we revealed differential synaptic inputs and intrinsic properties linked to the birthdate-related recruitment of V2a reticulospinal neurons.

## Spinal projections from individual hindbrain V2a neurons are organized based on their respective birthdates and innervate functionally matched spinal groups

Now that we had shown birthdate-related recruitment and its potential mechanisms in hindbrain V2a reticulospinal neurons, we examined their spinal projections in detail by labeling individual cells in each age group with dye electroporation in a series of transgenic lines labeling specific spinal neurons (*Figure 8*). We found distinct projection patterns across age groups at a single-cell level in a manner consistent with the overall age-related projection pattern of hindbrain V2a descending neurons, as revealed by photoconversion.

The early-born dorsal group (n = 8, position: 0.13–0.19) sent relatively thick axons through the medial longitudinal fasciculus and exhibited extensive axon collaterals throughout the spinal cord (*Figure 8B* ii). All the cells in this group had clear puncta-like structures - putative presynaptic terminals - in close proximity to the somata of PMNs (*Figure 8B* iv, 8 out of 8 cells) and SMNs (*Figure 8B* v, 4 out of 4 cells) but not spinal V2a neurons (*Figure 8B* vi, 0 out of 7 cells). Their collaterals were in the medial part of V2a neuropil (*Figure 8B* viii, ix) and clearly separated from the cell bodies of MCoDs located in the lateral part of the neuropil (*Figure 8B* vii vii', 0 out of 4 cells).

All the cells in the intermediate age group (n = 24, position: 0.32–0.48) had puncta-like structures on the ventrally located SMNs (*Figure 8C* v, 9 out of 9 cells) but only a third of the cells had puncta on the dorsally located PMNs (*Figure 8C* iv, 8 out of 24 cells). On the other hand, there was clear somatic innervation to the spinal V2a neurons in most of the cells (*Figure 8C* vi, 10 out of 13 cells).

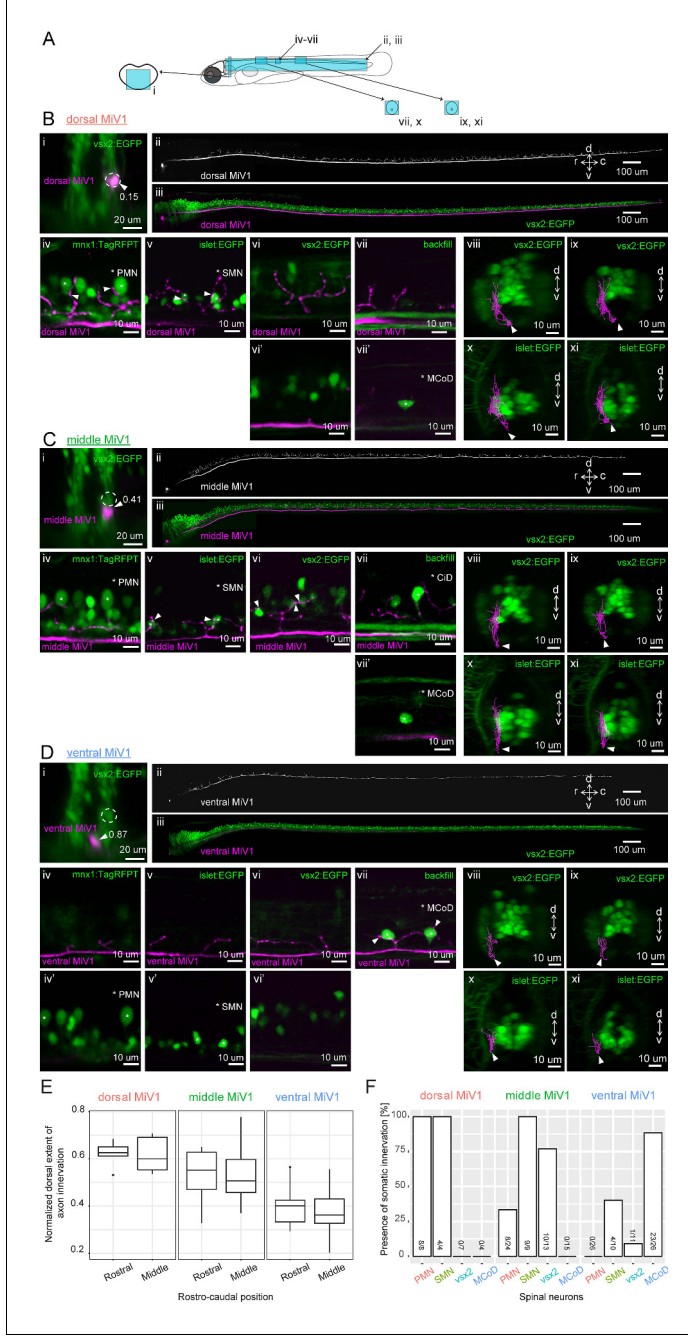

**Figure 8.** Spinal projections of V2a reticulospinal neurons in rhombomere 4. (**A**) Regions displayed (cyan patches) in B, C and D. (**B**) Spinal projection of an early-born V2a neuron (dorsal MiV1). (**i**) Coronal view of a dorsal MiV1 neuron (white arrowhead) electroporated with a dextran dye (magenta) in the background of V2a neurons (green). Dotted circles indicate dorsal MiV1. The number near the white arrowhead indicates normalized dorso-ventral position of the labeled cell (see Materials and methods). (**ii**) Side view of the spinal projection from dorsal MiV1. (**iii**) Same as in (**ii**) but overlaid on V2a neurons shown in green. (**iv-vii**) Sagittal optical slices showing the projection (**iv**) Projection relative to spinal mnx1+ neurons. Primary motoneurons (PMN) are indicated with asterisks. Processes juxtaposed to the cell body of PMN are highlighted with filled arrows. (**v**) Projection relative to spinal islet1+ neurons. Secondary motoneurons (SMN) are indicated with asterisks. Processes close to the cell body of SMN are highlighted with filled arrows. (**vi-vi'**) Projection relative to spinal V2a neurons. (**vi**) Optical slice showing the processes. (**vi'**) Optical slice medial to vi. (**vii-vii'**) Projection relative to a backfilled multipolar commissural descending neuron (MCoD). (**vii**) Optical slice showing the projection. (**vii'**) Optical slice lateral to vii showing the location of MCoD. (**viii-xi**) Coronal views of the projection of dorsal MiV1 in the rostral spinal cord (**viii, x**) and in

*Figure 8 continued on next page*

*Figure 8 continued*

the middle spinal cord (**ix, xi**) with V2a neurons labeled in EGFP (**viii, ix**) or with islet+ neurons labeled in EGFP (**x, xi**). (**C**) Spinal projection of an intermediate V2a neuron (middle MiV1). Images are organized similarly to B. (**vii**) Circumferential ipsilateral descending neuron (CiD) is indicated with asterisks. (**D**) Spinal projection of a late-born V2a neuron (ventral MiV1). Images are organized similarly to B. (**iv-iv'**) Projection relative to spinal mnx+ neurons. (**iv**) Optical slice showing the processes. (**iv'**) Optical slice medial to iv. (**v–v'**) Projection relative to spinal islet + neurons. (**v**) Optical slice showing the processes. (**v'**) Optical slice medial to v. (**vi-vi'**) Projection relative to spinal V2a neurons. (**vi**) Optical slice showing the processes. (**vi'**) Optical slice medial to v. (**vii**) Projection relative to a backfilled MCoD. (**vii-xi**) Coronal views are organized as in B. (**E**) Dorsal extent of axon innervation of V2a reticulospinal neurons in rhombomere 4. The dorsal extent of axon innervation is normalized to the thickness of spinal cord in dorsoventral axis. The main effect of hindbrain cell type (dorsal, middle, and ventral MiV1) was significant (p<0.001). (**F**) Somatic innervations of rhombomere 4 V2a reticulospinal neurons to spinal cell types. Percentage of cells showing somatic innervation to specific spinal cell types (PMN, SMN, vsx2, MCoD) are shown. The number of neurons showing innervation to a given class and the number of neurons examined are indicated at the bottom of each bar. The main effect of hindbrain cell type (dorsal, middle and ventral) was significant (p<0.001). The interaction of hindbrain cell type and spinal cell type was also significant (p<0.001).

DOI: https://doi.org/10.7554/eLife.42135.023

The following source data is available for figure 8:

**Source data 1.** Contains numerical data plotted in *Figure 8E*.

DOI: https://doi.org/10.7554/eLife.42135.024

---

Again, their collaterals were in the medial part of the neuropil (*Figure 8C* viii, ix) and clearly separated from the laterally displaced MCoDs (*Figure 8C* vii, vii', 0 out of 15 cells).

Axon collaterals from the late-born ventral group (n = 26, position: 0.68–0.92) were sparse (*Figure 8D* ii) and located mostly in the ventral part of the neuropil (*Figure 8D* viii-xi). The main axon tract was in the lateral part of the neuropil where no cell bodies of motoneurons and spinal V2a neurons are present (*Figure 8D* iv-vi). Most of them showed clear somatic innervations to MCoDs (*Figure 8D* vii, 23 out of 26 cells) but far fewer or no innervations to the soma of motoneurons and spinal v2a neurons (*Figure 8D* iv-vi, 0 out of 26 for PMN, 4 out of 10 for SMN and 1 out of 11 for spinal V2a neuron).

At the population level, there were statistically significant differences in the dorsal extents of spinal axon arborizations across age groups (*Figure 8E*, p<0.001). The axon arborization of each age group was also localized in a distinct medio-lateral position in the neuropil, as evident from coronal views (e.g. *Figure 8B* x-xi and 8D x-xi) and from their axon arborizations relative to the laterally displaced MCoDs (panels vii in *Figure 8B, C and D*). This is also in accordance with the chronological layering of V2a spinal projection revealed by photoconversion. These differences in axon arborization were reflected in the somatic innervation patterns to spinal neurons (*Figure 8F*). The interaction between MiV1 age groups and spinal cell types was statistically significant (p<0.001), meaning that each age group had distinct somatic innervations based on the spinal cell types. Indeed, the dorsal early-born group was the only group that consistently innervated the cell bodies of early-born motoneurons, PMNs, whereas the ventral late-born group was the only group that had somatic innervation to late-born interneurons, MCoDs. Thus, our results provided cellular-level anatomical evidence that hindbrain V2a neurons form parallel spinal pathways arranged based on birthdate and set the stage for electrophysiological analysis of these connections.

## Chronologically organized parallel synaptic connections to spinal neurons with distinct biophysical properties

Population- and cellular-level anatomy of hindbrain V2a neurons suggests that they exhibit distinct innervation patterns based on their time of differentiation: early-born neurons innervate early-born spinal neurons such as PMN while late-born neurons innervate late-born spinal neurons such as MCoD. To confirm these putative synaptic connectivity patterns and examine their properties in detail, we performed a series of paired whole-cell recordings from hindbrain descending neurons and spinal cord neurons (*Figure 9*).

First, we examined the synaptic connectivity of dorsal MiV1 (n = 22, position: 0.12–0.17), the early-born V2a descending neurons in rhombomere 4, to PMN (*Figure 9A*; n = 14) and MCoD

(*Figure 9B*; n = 8). The firing of an action potential in dorsal MiV1 led to a postsynaptic potential in PMN in most pairs (*Figure 10A*; 13 out of 14). In eight cases, the EPSPs had two clear components: a short latency component and a relatively longer latency component (*Figure 9A* iii). We tested the monosynaptic nature of the connections by raising magnesium and calcium concentrations to minimize the contribution of polysynaptic pathways (*Figure 9A* iii). In these cases (n = 7), the responses persisted, suggesting that they were mediated by monosynaptic connections. The longer latency component was blocked by a mixture of glutamatergic blockers, leaving only the shorter latency component (*Figure 9A* iii; n = 7). The longer latency component was also variable across trials while the short latency component was stable across trials. Therefore, the initial short latency response was probably due to an electrical synapse, and the later response that followed this was due to a glutamatergic synapse. On the other hand, none of the MCoDs showed a postsynaptic response to an action potential elicited in the dorsal MiV1 (*Figure 9B*; 8 out of 8). Taken together, our results indicate that in a manner consistent with their spinal projection patterns, dorsal MiV1s, the early-born V2a neurons in rhombomere 4, show selective synaptic connectivity to PMNs, the early-born spinal neurons.

In contrast, ventral MiV1s (n = 29, position: 0.67–0.91), the late-born V2a descending neurons in rhombomere 4, showed the opposite pattern of synaptic connectivity. None of the PMNs showed a postsynaptic response to an action potential elicited in ventral MiV1 (*Figure 9C*; 0 out of 8), while one-third of MCoDs examined showed a clear postsynaptic potential (*Figure 9D*; 8 out of 21). The responses in these MCoDs persisted in high magnesium and calcium solution, suggesting they were generated monosynaptically. In four out of eight cases, we observed clear biphasic responses (*Figure 9D* iii), the latter component of which was blocked by glutamatergic blockers (*Figure 9D* iii; n = 4). The amplitude of the short latency component was stable across trials in contrast to the later component mediated by glutamatergic synapses, suggesting that the early component was probably due to electrical synapses. Thus, in a manner consistent with their spinal projection patterns, ventral MiV1s showed selective monosynaptic connectivity to MCoDs, the late-born spinal interneurons that are only active during slow swimming (*McLean et al., 2008*).

To see if other V2a descending neurons and non-V2a descending neurons show similar connectivity patterns based on time of differentiation, we also looked at the connectivity of early-born V2a neurons in rhombomere 6, MiD3i, (*Figure 9E,F*) and the earliest born descending neurons in rhombomere 4, the Mauthner cell (M-cell), (*Figure 9G,H*) to PMNs and MCoDs. In seven out of seven cases, a clear biphasic postsynaptic potential was observed in PMNs in response to an action potential in MiD3i (*Figure 9E*). This response persisted in high magnesium and calcium solution, suggesting that they were monosynaptic. Once again, the early component was stable across trials while the late component was variable and blocked by glutamatergic blockers (*Figure 9E* iii; n = 7), suggesting that they were electrical and glutamatergic, respectively. The M-cell also showed similar, but much stronger connections. In 14 out of 14 cases, PMNs showed very strong postsynaptic potentials leading to spiking activity in response to a single action potential in the M-cell (*Figure 9G* iii). These responses also persisted in high magnesium and calcium solution, suggesting that the connections were monosynaptic. These responses were completely blocked by a cholinergic blocker (*Figure 9G* iii; n = 10), indicating that they were mediated by cholinergic synapses, in accordance with existing literature (*Koyama et al., 2011*; *Clemente et al., 2004*). MCoDs on the other hand showed a long latency complex postsynaptic potential in response to one M-cell action potential, suggesting that the connection is probably polysynaptic (*Figure 9H* iii; 13 out of 13 cases). Most, if not all, of the responses disappeared in high magnesium and calcium solution, further supporting that they are mediated polysynaptically (*Figure 9H* iii; n = 10). In sum, our findings indicate that both types of early-born descending neurons make synaptic connections to the early-born PMNs but not to the late-born MCoDs.

Overall, the connection probability was statistically different depending on the hindbrain cell type and the spinal cord cell type (*Figure 9I*; main effect of hindbrain cell type, p<0.001; main effect of spinal cell type, p<0.0001). In fact, PMNs and MCoDs received synaptic inputs from completely different sets of reticulospinal neurons, with each set corresponding to distinct age group (*Figure 9I*). Thus, these results showed electrophysiologically that early-born and late-born hindbrain neurons form parallel pathways to the spinal cord and these pathways are organized based on the time of differentiation, supporting the idea that parallel hindbrain-spinal cord circuits emerge in sequence during development.

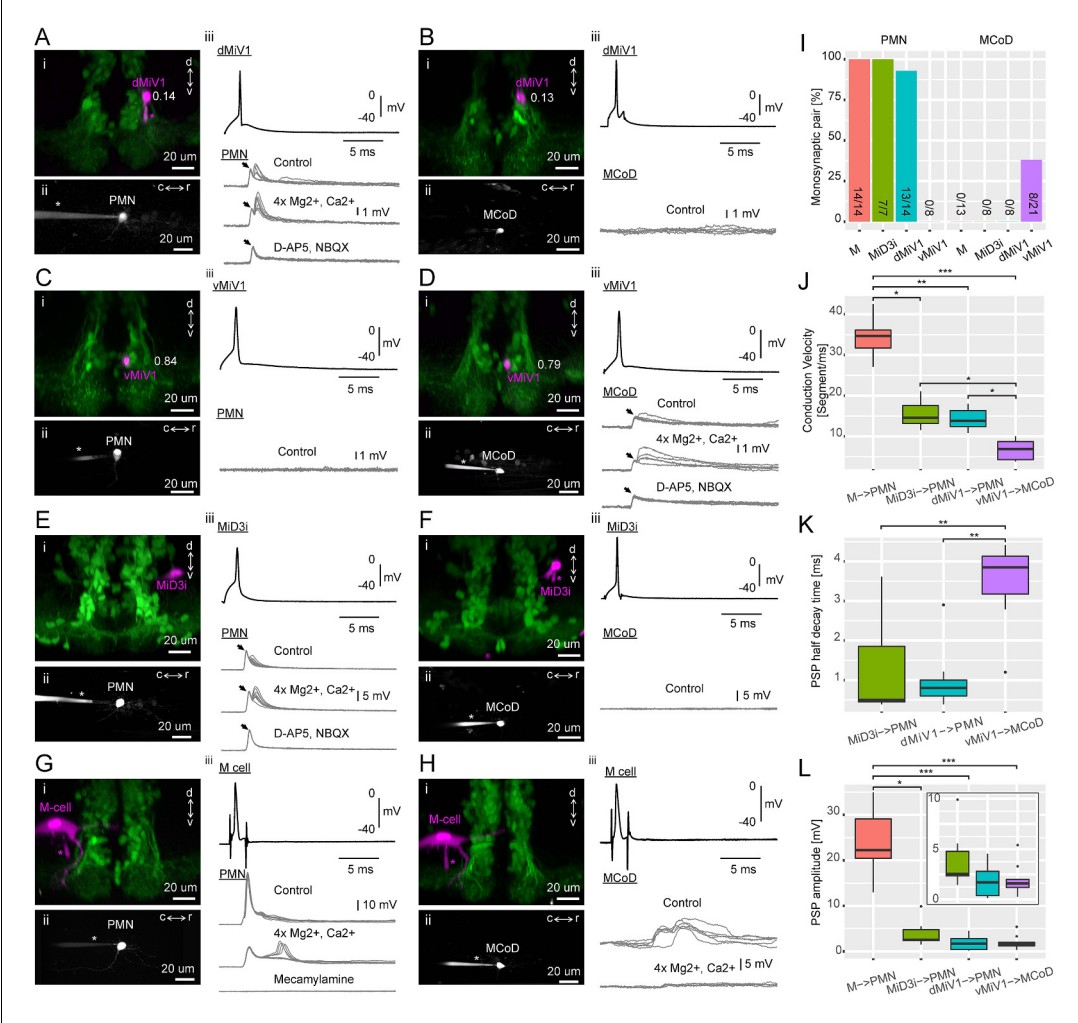

**Figure 9.** Synaptic connectivity of hindbrain V2a descending neurons. (**A–H**) Paired recordings of hindbrain neurons and spinal neurons. dMiV1, dorsal MiV1; vMiV1, ventral MiV1; M, Mauthner; PMN, primary motoneuron; MCoD, multipolar commissural neuron. (**i**) Coronal view of a patched hindbrain neuron (magenta) overlaid on V2a neurons (green). An asterisk indicates the patch pipette in use. The numbers in A-D indicate normalized dorsoventral position of patched MiV1 neurons. (**ii**) Side view of a patched spinal neuron (gray). (**iii**) Traces of postsynaptic potentials (PSPs) in a spinal neuron following spikes generated in a hindbrain neuron in control condition (top gray trace), in high divalent cation solution (middle gray trace) and following the blockade of chemical synapses (bottom gray trace). A black arrow highlights the putative electrical component of the postsynaptic potential. For simplicity, only one action potential trace is shown (top black trace). (**I**) Monosynaptic connections of hindbrain descending neurons to spinal neurons. Percentage of pairs showing monosynaptic connections to spinal neurons are shown. The number of connected pairs and the number of pairs examined are indicated at the bottom of each bar. The main effect of hindbrain cell types on the percentage of monosynaptic connection was significant ($p < 0.001$). The main effect of spinal cell types was also significant ($p < 0.001$). (**J**) Conduction velocity of monosynaptically connected pairs. Conduction velocity was defined as the number of muscle segments an action potential propagated in a millisecond. Box and whisker plots represent median (horizontal line across a box) as well as first and third quartiles. Asterisks mark significant differences between given connections (Dunn's test, *$p < 0.05$, **$p < 0.01$, ***$p < 0.001$). (**K**) Half decay time of monosynaptic PSPs. Data are presented as in J. (**L**) Amplitude of monosynaptic PSPs. Data are presented as J. (inset) Zoomed-in box and whisker plots for the pairs with smaller PSP.

DOI: https://doi.org/10.7554/eLife.42135.025

The following source data is available for figure 9:

**Source data 1.** Contains numerical data plotted in *Figure 9J*.
DOI: https://doi.org/10.7554/eLife.42135.026
**Source data 2.** Contains numerical data plotted in *Figure 9K*.
DOI: https://doi.org/10.7554/eLife.42135.027
**Source data 3.** Contains numerical data plotted in *Figure 9L*.
DOI: https://doi.org/10.7554/eLife.42135.028

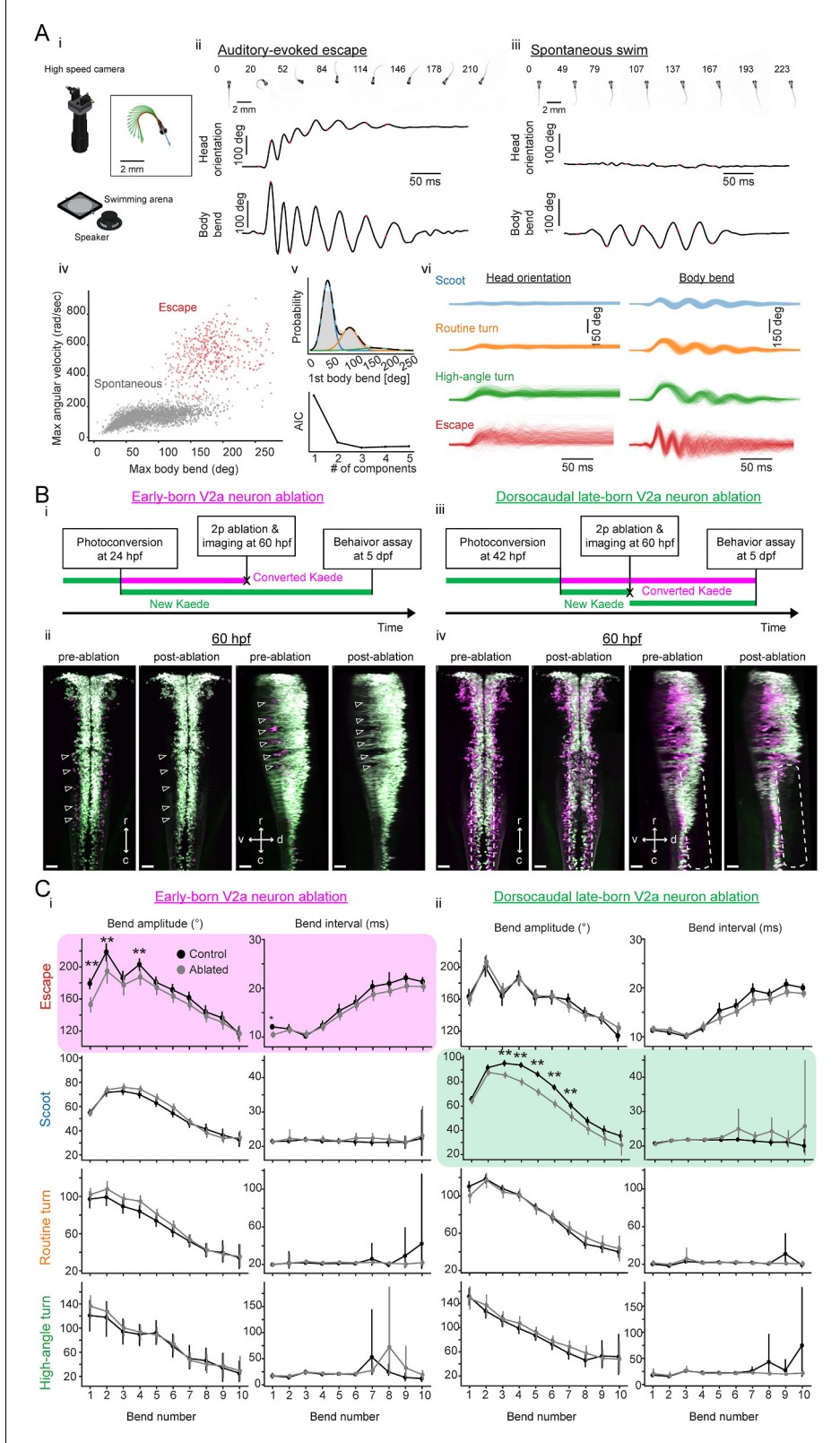

**Figure 10.** Contributions of hindbrain V2a neurons to evoked and spontaneous locomotion. (**A**) Examined locomotion patterns. (**i**) Experimental setup. (inset) Example fish image showing tracking of head orientation and tail curvature (see Materials and methods). Blue arrow, head orientation; Red dots, tracked midline points; Green arrows, tangent vectors along the midline points. (**ii**) Auditory-evoked escape response. (Top row) Images of fish

*Figure 10 continued on next page*

*Figure 10 continued*

during escape response. The number on each the images is time from the stimulus onset in milliseconds. (Middle row) Time course of head orientation. (Bottom row) Time course of total body bend (see Materials and methods). (iii) Spontaneous swimming. Panels are organized similarly to ii but the number on each image is time from the movement onset in milliseconds. (iv) Scatter plot of maximum bend amplitude (during a swim episode) and maximum angular velocity for auditory-evoked escapes (n = 426 episodes, N = 10 fish, red points) and spontaneous swims (n = 6088 episodes, N = 15 fish, gray points). (v) (Top) Probability of initial bend amplitude for 6056 spontaneous swim events. A fit of gaussian mixture model identified three components. (Bottom) The Akaike information criterion (AIC) as a function of number of components. Three components led to the lowest AIC value. (vi) Overlaid traces of head orientation and total body bend for three identified spontaneous swim categories (Scoot, 3969 episodes, blue; Routine turn, 1565 episodes, orange; High-angle turn, 376 episodes, green) and auditory-evoked escape (Escapes, 381 episodes, red). (B) Femtosecond laser ablation of hindbrain V2a neurons. (i) Experimental procedure for the ablation of early-born V2a neurons. (ii) Maximum intensity projections before and after ablation at 60 hpf (left, dorsal view; right, side view). White open arrowheads indicate the locations of the early-born V2a neurons.(iii) Experimental procedure for the ablation of dorsocaudal late-born V2a neurons. (iv) Maximum intensity projections before and after ablation at 60 hpf (left, dorsal view; right, side view). White rounded rectangles indicate the location of the late-born dorsocaudal V2a neurons. r, rostral; c, caudal; d, dorsal; v ventral; scale bars, 30 μm. (C) Effects of hindbrain V2a neuron ablations on the bend amplitudes and periods (or intervals) of the four distinct locomotion patterns. (i) Effects of the early-born V2a neuron ablation. Bend amplitude and interval are quantified bend-by-bend from 1st bend to 10th bend (see Materials and methods). Asterisks mark significant differences between ablated and control fish (**p<0.01, corrected for multiple comparisons, Holm test). Error bars, 99 percent confidence interval. The swim category that showed significant effects are highlighted in magenta. (ii) Effects of the late-born dorsocaudal V2a neuron ablation. Panels are organized as in i. The swim category that showed significant effects are highlighted in green.

DOI: https://doi.org/10.7554/eLife.42135.029

The following source data and figure supplements are available for figure 10:

**Source data 1.** Contains numerical data plotted in *Figure 10C*.
DOI: https://doi.org/10.7554/eLife.42135.035

**Figure supplement 1.** Confirmation of hindbrain V2a neuron ablations at four dpf.
DOI: https://doi.org/10.7554/eLife.42135.030

**Figure supplement 2.** Effects of age-specific ablations of hindbrain V2a neurons on global swim parameters.
DOI: https://doi.org/10.7554/eLife.42135.031

**Figure supplement 3.** Criteria used to limit analyses of body bend amplitudes and periods to the first 10 bends of an episode for all swim categories.
DOI: https://doi.org/10.7554/eLife.42135.032

**Figure supplement 4.** Probability distributions of the first body bend amplitude of spontaneous swim events in the ablated and control groups.
DOI: https://doi.org/10.7554/eLife.42135.033

**Figure supplement 5.** Overlaid traces of body bend during scoot in the ablated and control groups.
DOI: https://doi.org/10.7554/eLife.42135.034

We then examined biophysical properties of these synaptic connections. We first examined differences in conduction velocity among connected pairs (*Figure 9J*): the youngest ventral MiV1/MCoD pairs were the slowest among the connected pairs. The middle-aged pairs (dorsal MiV1/PMN and Mid3i/PMN) were slower than the oldest M-cell/PMN pairs. We also examined differences in the time constants of synaptic potentials, excluding the M-cell/PMN pair that showed spiking activity (*Figure 9K*). The youngest ventral MiV1/MCoD pairs showed significantly longer time constants than the older V2a neurons (*Figure 9K*). With respect to the amplitudes of the synaptic potentials, the M-cell/PMN pair was the only pair that was statistically different from other pairs (*Figure 9L*). However, consistent with previous studies (*McLean et al., 2007*; *Menelaou and McLean, 2012*), the early-born PMNs and the late-born MCoDs examined here differed significantly in input resistance (92.3 ± 18.2 M Ohm and 514 ± 172 M Ohm, respectively; Wilcoxon test, p<0.001), suggesting that the underlying synaptic currents were systematically stronger for the early-born hindbrain neurons than the late-born hindbrain neurons. Altogether, we found systematic differences in conduction velocities, synaptic decay times and synaptic currents between the two parallel pathways: the early-born pathway is equipped with fast conduction velocities, fast time constants of synaptic potentials and stronger synaptic currents while the late-born pathway is equipped with slow conduction

velocities, slow time constants of synaptic potentials and weaker synaptic currents. The differences in the biophysical properties as well as connectivity patterns are consistent with the distinct types of axial motor activity observed during the locomotor patterns in which these pathways participate. The net effects of the distinct biophysical properties and connectivity patterns associated with the early- and late-born pathways is that the former could more quickly bring the PMNs throughout the spinal cord to their firing thresholds, whereas the latter could more slowly bring the SMNs in the caudal spinal cord to their firing thresholds. As a consequence, activation of early-born pathways could produce more powerful whole body bending observed during different crude forms of locomotion (escape and struggle)(*McLean et al., 2008*; *Liao and Fetcho, 2008*), whereas activation of late-born pathways could produce weaker bending of the caudal tail region with a long rostrocaudal delay, as observed during the refined locomotion (*McLean et al., 2008*). In sum, these results suggest that the parallel hindbrain-spinal cord circuits that develop in sequence and have distinct connectivity patterns and biophysical properties contribute to the distinct locomotor patterns that zebrafish display in sequence.

## Behavioral contributions of distinct age groups of hindbrain V2a descending neurons

Systematic characterization of the development, recruitment, and connectivity of V2a descending neurons led us to hypothesize that early-born hindbrain V2a descending neurons contribute to the powerful whole-body movements observed during the stimulus-evoked locomotion that develops early in development while their late-born counterparts contribute to the weaker tail-restricted movements observed during the spontaneous locomotion that develops later. To test this hypothesis, we separately ablated either the early-born or the late-born hindbrain V2a descending neurons and at 5 days post-fertilization (dpf) examined the effects of each of these ablations on a few distinct patterns of locomotion (*Figure 10*).

To examine a wide range of swimming patterns in the freely swimming condition, we employed two experimental conditions: auditory-evoked escape responses (*Figure 10A* ii) and spontaneous swimming (*Figure 10A* iii). In each condition, we monitored movements of the fish using a high-speed camera (*Figure 10A* i). Auditory stimulus-elicited escape episodes consisted of large and fast changes in head orientation as well as body bends (*Figure 10A* ii), these being characteristic of the crude locomotion that develops early. On the other hand, spontaneous swim episodes mostly consisted of small changes in head orientation, and slow and small bends restricted to the caudal portion of the tail (*Figure 10A* iii), and these are typical of the more refined locomotion that develops later. The distinctiveness of these two swim types became quite apparent when for each swim episode the maximum angular velocity (see Materials and methods) was plotted against the maximum body bend amplitude (*Figure 10A* iv). In this plot, the points corresponding to escapes and slow swims clearly segregated into distinct clusters. Consistent with the findings of a previous study (*Burgess and Granato, 2007*), when we looked at the distribution of the amplitude of the first total body bend in each spontaneous swim episode we found it to be multimodal, suggesting that spontaneous swim episodes consisted of distinct subtypes (*Figure 10A* v). A Gaussian mixture model fit indicated that the best fit to the distribution was using a linear combination of three distinct gaussian distributions, suggesting that there were potentially three subcategories of spontaneous swim episodes (see Materials and methods). The timeseries of body bends and head orientation in each category showed consistent motor patterns (*Figure 10A* vi), further supporting the idea that fish at 5 dpf exhibit at least three distinct motor patterns spontaneously. We referred to these patterns as 'scoot', 'routine turn' and 'high-angle turn' based on the nomenclature adopted in previous studies (*Burgess and Granato, 2007*; *Marques et al., 2018*). Combined with auditory-evoked escape responses, we examined four distinct motor patterns in total (*Figure 10A* vi).

We focused on two age groups of hindbrain V2a descending neurons based on their times of differentiation and their recruitment patterns. The first group consisted of neurons born before 24 hpf (10 fish for the ablated group; 10 fish for the control group). We targeted all these neurons in the hindbrain as the optical backfill and Ca$^{2+}$ imaging indicated that all of them were descending neurons involved in strong locomotion. We visualized these neurons by photoconverting Tg(vsx2:Kaede) at 24 hpf, and at 60 hpf we ablated these neurons by femtosecond laser pulses (*Figure 10B* i; 131 ± 17.3 cells/fish). We delivered multiple pulses to each neuron (typically 5–10 pulses/neuron) until the fluorescence signal suddenly decreased. Neurons labeled with photoconverted Kaede

disappeared with no clear unintended damage to the nearby cells and processes (*Figure 10B* ii). To confirm that these neurons did not recover before the behavior experiments, we imaged them again at four dpf (*Figure 10—figure supplement 1A*). Comparison with the control group at 4 dpf clearly showed that these early-born neurons did not recover from ablation. The second group of neurons targeted for ablation consisted of neurons born from 42 hpf to 60 hpf (*Figure 10B* iii; 10 fish for the ablated group; 10 fish for the control group). We focused on the neurons in the dorsocaudal hindbrain because the optical backfill and $Ca^{2+}$ imaging indicated these neurons are descending neurons involved in spontaneous slow locomotion. We identified these V2a neurons based on the lack of photoconverted Kaede at 60 hpf after the photoconversion at 42 hpf (*Figure 10B* iii) and their relative position to the nearby early-born neurons (508 ± 15 cells/fish; see Materials and methods). Femtosecond laser pulses ablated these neurons with no clear unintended damage to the nearby early-born neurons (*Figure 10B* iv). Comparison with the control group at 4 dpf clearly showed the reduction of the late-born neurons in the dorsocaudal hindbrain (*Figure 10—figure supplement 1B*).

To assess if ablations had resulted in global changes in swim episodes, we examined total swim distance per episode, maximum swim velocity per episode, mean swim velocity per episode, episode duration, and total number of bends per episode (see Materials and methods) for each of the four distinct swim types (*Figure 10—figure supplement 2*). We found that for scoots all of the foregoing global parameters were significantly reduced in the late-born V2a neuron ablation group (*Figure 10—figure supplement 2B*, p<0.01, corrected for multiple comparisons, Holm test). For routine turns, these ablations produced a slight reduction in swim distance and mean swim velocity, but at a lower significance level (*Figure 10—figure supplement 2B*, p<0.05, corrected for multiple comparisons, Holm test). This is consistent with our prediction that the late-born V2a neuron contributes to the weaker tail-restricted movements observed during spontaneous swims. On the other hand, ablation of early-born V2a neurons did not result in changes in any of these global parameters. Neither ablation of the early-born or the late-born neurons resulted in a change in the onset latency of escape (*Figure 10—figure supplement 2A*). The lack of phenotype for the early-born group ablation is perhaps not surprising as the recruitment and connectivity of the early-born V2a neurons suggest that the this group is likely to contribute most to the powerful body bends observed during the initial phase of escapes.

To examine the effect of ablations on a bend-by-bend basis, for each swim type, we compared the amplitudes and durations of body bends between ablated and matched control groups (*Figure 10C*). This analysis is based on the assumption that each swim type is so stereotypical that one could compare identically-numbered bends across episodes. We assess this assumption based on the timing of each bend with respect to the onset of the episode (*Figure 10—figure supplement 3*): we examined how discriminable the time of maximum amplitude of each bend is from that of the preceding bend (see Materilas and methods). We found that the discriminability during escapes dropped suddenly after the sixth bend but for other swim types discriminability decreased more gradually until the tenth bend which marked the end to most of these episodes (*Figure 10—figure supplement 3B* ii). Based on this observation, we decided to perform bend-by-bend analysis up to the 10th bend across all swim types for consistency. Relative to the control group, the early-born V2a ablation group exhibited significant decreases in the amplitudes of the first, second and fourth bends, as well as a decrease in the duration of the first bend of escape (*Figure 10C* i, Escape; p<0.01, corrected for multiple comparisons, Holm test). However, we observed no statistically significant changes in the three other slower locomotor patterns (*Figure 10C* i, Scoot, R-turn and High-angle turn). On the other hand, in the late-born V2a ablation group we saw no clear changes in the escape response (*Figure 10C* ii, Escape). Instead, this group showed a decrease in the amplitudes of bends in the later phase of scoot (*Figure 10C* ii, Scoot; p<0.01, corrected for multiple comparisons) but not the other two spontaneous swim patterns involving turning (*Figure 10C* ii, Routine turn and High-angle turn). These observations comport well with the recruitment pattern and the spinal projections we observed earlier for the neurons we ablated here. Furthermore, the fact that we observed significant changes only in the late phase of scoots suggests that late-born V2a neurons may play a role in maintaining the excitatory drive in the spinal cord during slow forward locomotion.

To gain assurance that we compared the same set of spontaneous locomotor behaviors between the control and ablated groups, we compared the distributions of the first body bend we used to subcategorize spontaneous locomotor behavior (*Figure 10—figure supplement 4*). Although there

were slight differences in the peak amplitudes of the probability distributions across the groups, the overall shapes of the distributions were consistent, indicating that we compared the same sets of locomotor behaviors across the control and ablated groups. To examine this more closely for scoot behavior, which showed significant deficits upon ablation of late-born caudal hindbrain neurons, we examined the time course of the amplitude of body bending during this behavior (*Figure 10—figure supplement 5*). Traces from all the episodes of scoot from the ablated fish (*Figure 10—figure supplement 5*, orange) and the control fish (*Figure 10—figure supplement 3*, blue) are overlaid for each ablation group. In both ablation groups, the ablated and control groups both exhibited the small amplitude body bends with slow left-right alternation (*Figure 10—figure supplement 5*) that is characteristic of scoot behavior (*Figure 10A* vi). Furthermore, the overlaid traces of the late-born V2a ablation and control groups clearly revealed that the bend amplitudes of scoots decreased in the later phase of scoot in the ablated group (*Figure 10—figure supplement 5A*). This further strengthened the idea that late-born V2a neurons of the caudal hindbrain play a role in the maintenance of excitatory drive during slow forward locomotion. To summarize, our ablations did not change the set of distinct locomotor behaviors fish exhibit spontaneously.

While examining the results of ablations, we noticed that the early-born and the late-born control groups exhibited significantly different bend amplitudes during scoot (*Figure 10C* i, Scoot; *Figure 10C* ii, Scoot, p<0.01). Comparison to an independent control group that was not subjected to photoconversion suggested that photoconversion at 42 hpf impacted the kinematics of scoot at five dpf (data not shown). Although this does not undermine the observed significant decrease in bend amplitudes of scoot in the late-born ablation group, it does raise some concerns about the lack of significant effects in the early-born ablation group. For instance, it is conceivable that ablation of early-born neurons does not reveal a decrease in scoot bend amplitudes because of a floor effect, wherein the observed bend amplitudes somehow represent the lowest values that the fish can inherently display in order to produce a scoot (*Figure 10C* i, Scoot). However, we think this is unlikely because we noticed a slight increase rather than decrease in bend amplitudes after ablation of the early-born neurons (*Figure 10C* i, Scoot). In any case, the effects of photoconversion itself further justifies our use of matched control for each ablation group.

In conclusion, with ablation experiments we confirmed our hypothesis about the functional roles of early-born and late-born V2a neurons in freely swimming fish. Moreover, the double dissociation of the two locomotor patterns indicates that the underlying V2a descending pathways are largely independent, in accordance with their parallel circuit arrangement resulting from the layered growth of their spinal projections. This suggests that the layered development of new connections in the nervous system underlies its ability to integrate new circuits into existing circuitry in such a way as to allow the different circuits to operate autonomously. Interestingly, previous studies showed that the ablation of ventral reticulospinal neurons, the hindbrain V2a neurons we classified as the intermediate-aged group, affects slow but large turns with no clear kinematic effects on forward locomotion and suggested that the descending pathway for turning behaviors is independent of the one for forward locomotion (*Huang et al., 2013*; *Orger et al., 2008*). Collectively, this suggests that the array of locomotor behaviors fish display after birth are supported by sequentially developing parallel hindbrain-spinal cord circuits, with each circuit accounting for a specific range of kinematics and dynamics through its unique connectivity pattern and biophysical properties.

## Discussion

We examined hindbrain V2a neurons and their descending pathways and uncovered a chronology-based architecture that explains the postnatal development of locomotor behaviors (*Figure 11*). Hindbrain V2a descending neurons born early are also the first to project to the spinal cord where they innervate similarly early-born motoneurons throughout the spinal cord with fast conducting axons and synapses that exhibit fast dynamics. Consistent with their connectivity pattern and biophysical properties, they contribute to the crude forms of locomotion that consist of powerful bends of the whole-body. Hindbrain V2a descending neurons born later give rise to a more laterally located layer of descending pathways that function in parallel to existing pathways: through similarly late-born premotor neurons in the rostral spinal cord, these V2a neurons polysynaptically connect to caudal motoneurons via slow conducting axons and synapses that exhibit slow dynamics. Consistent with the connectivity pattern and biophysical properties, they contribute to the more refined and

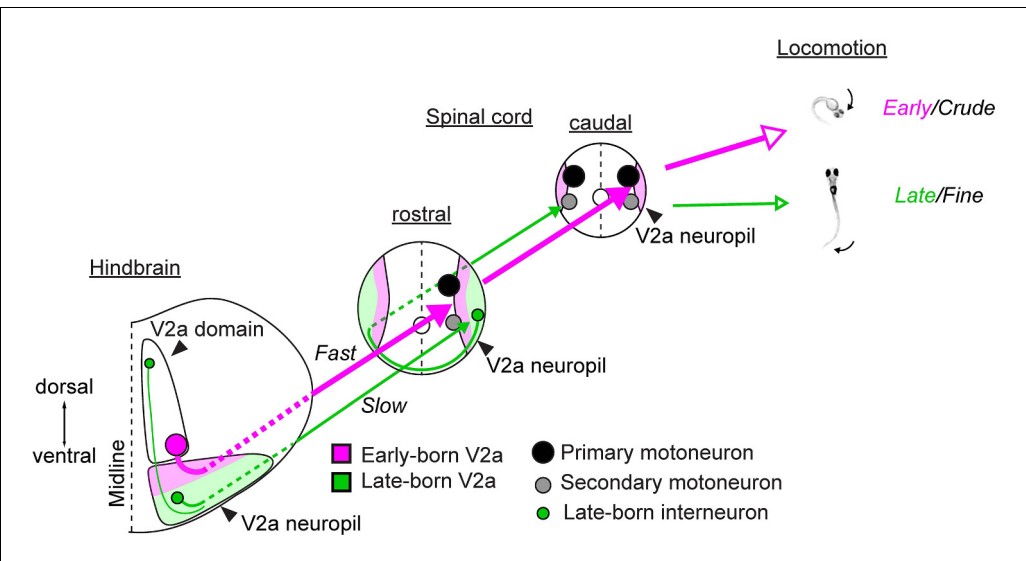

**Figure 11.** Chronological architecture of hindbrain V2a descending pathways that underlies diversification and increase in sophistication of zebrafish locomotor behaviors. A schema illustrating the relationship between the birthdate-related organization of hindbrain V2a descending pathways and the development of locomotor behaviors. Cell bodies are represented in circles. A larger circle means lower excitability. Arrows indicate descending axons. A larger arrow indicates faster and stronger connection. Neuropil regions are color-coded to indicate the separation based on the time of differentiation. Magenta area, early-born neuropil; Green area, late-born neuropil. Open arrows indicate the correspondence between pathways and axial movements during locomotion. The size of open arrow indicates the strength of the movement.
DOI: https://doi.org/10.7554/eLife.42135.036

weaker locomotion that appears later and that consists of weaker bends of the caudal tail. In sum, our results reveal that the chronological layering of parallel circuits and the systematic variation in the connectivity patterns and biophysical properties of these circuits underlie the diversification and increase in sophistication of the locomotor repertoire.

Such chronology-based incorporation of new motor circuits in parallel to existing circuits may provide distinct advantages over a non-parallel organization such as exists within the midbrain descending pathways (*Wang and McLean, 2014*). First, it allows the motor patterns produced by these circuits to be controlled independently of one another. Such independent control may be critical for some behaviors such as prey capture, for instance, that require flexible coordination of multiple motor patterns. Indeed, during the final approach phase of prey capture, zebrafish can increase the speed of forward swimming to values approaching those seen during fast behaviors established earlier in development, but without displaying the large lateral head displacements seen in these earlier-established behaviors (*Patterson et al., 2013*). This suggest that fish can generate fast forward swimming by strongly driving only the late-born pathway that predominantly drives tail movements. Second, this parallel organization allows incorporation of new circuits controlling increasingly sophisticated motor patterns in an open-ended way. Interestingly, the development of the corticospinal pathway in mammals appears to fit into this view; in this pathway, spinal tracts are established postnatally in the lateral part of the spinal cord after the prenatal development of other descending pathways (*Lakke, 1997*; *KUYPERS, 1962*), and these tracts contribute to skilled forelimb movements that animals gradually develop postnatally (*Martin, 2005*; *Alstermark and Isa, 2012*). Further analysis will be necessary, however, to assess the degree to which the corticospinal pathway can function without contributions from other descending pathways (see *Esposito et al., 2014*). In sum, chronology-based parallel incorporation of new motor circuits provides certain advantages over non-parallel organization.

There is some evidence that suggests that such chronology-based parallel incorporation of new circuits extends beyond motor systems. In the lateral line system of zebrafish, sensory afferents in the hindbrain are topologically organized based on their birthdate (*Pujol-Martí et al., 2012*). In the

cerebellar molecular layer in mice, parallel fiber axons from granule cells line up in order of age (*Espinosa and Luo, 2008*). In the fly visual system, the growth cones of photoreceptor afferents are segregated based on their birth order (*Kulkarni et al., 2016*). Also, in the mouse hippocampus, age-matched subpopulations of principal neurons are interconnected (*Deguchi et al., 2011*). Although the link between age-dependent layered circuit organization and the development of behaviors remains to be demonstrated in these systems, given how widely this pattern is observed, we suspect that it is a fundamental organizing principle in the brain that allows for implementation of new circuits in parallel to existing ones to meet the demands of an increasingly diverse behavioral repertoire following an animal's birth.

The systematic changes in connectivity patterns and biophysical properties we observed in the chronologically layered parallel circuits fit well with the general development pattern of movements observed in many vertebrates (*Kagan and Herschkowitz, 2006*; *Harlow and Harlow, 1965*; *Fox, 1965*; *Drapeau et al., 2002*). For example, in humans, global and fast body movements such as startle develop first followed by isolated limb movements and fine motor skills (*de Vries et al., 1982*; *Gallahue et al., 2012*). Broad and direct connectivity patterns equipped with fast biophysical properties would be suitable for the global and fast body movements that develop early. Localized and indirect connectivity patterns equipped with slow biophysical properties would be suitable for the finer movements that develop later. It would be interesting to see if the neural pathways underlying such behaviors indeed exhibit systematic changes in connectivity patterns and biophysical properties.

The aforementioned systematic changes in biophysical properties and connectivity patterns may not be restricted to motor systems. Indeed, correlations between morphological and physiological properties and the time of differentiation have also been reported in the mammalian cortex (*Butt et al., 2005*; *Miyoshi et al., 2007*) and hippocampus (*Picardo et al., 2011*; *Marissal et al., 2012*). Although the computational and behavioral roles of these systematic changes in morphological and physiological properties remains to be answered, it seems likely that they represent a general strategy utilized by nervous systems to support the wide range of neural processes required for the complete behavioral repertoire of animals: from the fast and global sensorimotor processing observed in innate reflexive behaviors to the more sustained and finer processing presumed to happen in deliberate and sophisticated cognitive behaviors that animals acquire later.

The hindbrain contains a series of cell types that are arranged in columns that run rostrocaudally throughout the hindbrain (*Kinkhabwala et al., 2011*; *Koyama et al., 2011*; *Gray, 2013*). Each cell type exhibits a specific combination of transcription factor expression, neurotransmitter identity and axonal projection pattern. Hindbrain V2a neurons comprise one such cell type. As these cell types span across various sensorimotor circuits in the hindbrain, it has been proposed that each cell type may provide a fundamental neural operation that is broadly useful to many of these circuits (*Kinkhabwala et al., 2011*; *Koyama and Pujala, 2018*). Indeed, there is some evidence suggestive of the involvement of hindbrain V2a neurons in other hindbrain motor circuits besides locomotor circuits. In mice, genetic ablation of all V2a neurons in the nervous system produces deficits in respiration (*Crone et al., 2012*). A class of premotor neurons in the horizontal eye movement circuit partly corresponds to putative V2a neurons in larval zebrafish (*Lee et al., 2015*). These neurons show persistent activity correlated to eye position but the time constant of persistent activity varies across cells (*Miri et al., 2011a*). This raises the possibility that, as in locomotor movements, eye movements are controlled by a series of V2a neurons that show systematic changes in biophysical properties based on the time of differentiation. Perhaps one fundamental role of hindbrain V2a neurons is to provide to hindbrain motor circuits a series of excitatory drives that vary in duration so that these circuits can result in a behavioral repertoire containing movements with a wide range of temporal dynamics. Precise functional mappings and detailed circuit analyses will be necessary to examine if this is indeed the case.

Previous studies of spinal V2a neurons in larval zebrafish employed tail beat frequency as a readout of locomotor speed and revealed that distinct age groups are recruited as a function of locomotor speed: the early-born group is active during fast locomotion while the late-born group is active during slow locomotion (*Kimura et al., 2006*; *McLean et al., 2008*). This seems at odds with the recruitment of early-born hindbrain V2a descending neurons during struggles which generate powerful whole-body bends but at low tail beat frequencies. However, in the context of forward locomotion, the strength of body bends also increases as locomotor speed increases (*McLean et al., 2008*).

Thus, it is possible that spinal V2a neurons are recruited based on the strength of body bends similar to hindbrain V2a descending neurons. Consistent with this idea, the earlier-born spinal V2a neurons have longer axons than the later-born ones (*Menelaou et al., 2014*), suggesting that the early-born population contributes to large body bends through its innervation to motoneurons across many segments. Indeed, the activity of CiD interneurons, putative early-born V2a population, is enhanced when fish exhibit a larger initial bend during escape (*Bhatt et al., 2007*). Previous studies in tadpole also identified putative spinal V2a neurons recruited during struggles (*Li et al., 2007*). These neurons can also fire during fast beat frequencies (*Li et al., 2007*, *Figure 4E*; *Li, 2015*, *Figure 7A*), suggesting that they correspond to the early-born population active during fast swimming in zebrafish. Taken together, this suggests that like the hindbrain V2a population we examined, spinal V2a neurons may also exhibit recruitment pattern based on the strength of movements. However, direct examination of the early-born spinal V2a neurons during struggles will be necessary to fully resolve this issue.

A shared set of early-born hindbrain V2a descending neurons is recruited during two crude forms of locomotion, struggles and escapes. This suggests that the early-born V2a descending pathway is in use during both behaviors and generates strong activation of axial muscles along the whole body as observed in both behaviors. However, these behaviors are distinct behaviors: escape is forward locomotion with fast tail beat frequency while struggle is backward locomotion with slow tail beat frequency. How do these differences emerge? Within the early-born V2a descending neurons, majority of the cells show stronger activity during escapes. This suggests that the further recruitment of the early-born pathway underlies the transition from struggles to escapes. However, it is also possible that non-V2a descending neurons that were not examined in this study show activity only during struggles and recruit spinal neurons active only during struggles such as CoLA (*Liao and Fetcho, 2008*). Comprehensive functional and anatomical examination of all descending neurons will be necessary to understand how these two distinct behaviors are generated.

In principle, it is possible for nervous systems to acquire new patterns of activity corresponding to new behaviors through the development of neuromodulatory systems because neuromodulation permits a fixed complement of neurons to give rise to many different patterns of activity (*Bargmann, 2012*; *Marder, 2012*). Indeed, in larval zebrafish blocking dopamine signaling through D4 receptors leads to less frequent but longer swim episodes (*Lambert et al., 2012*). It remains to be seen if these episodes correspond to the crude form of swims observed in embryos (*McLean and Fetcho, 2009*) or to the refined form of swims but with a larger number of tail beats. Nevertheless, this indicates that D4 receptor signaling can either suppress or modify the late-born pathway we described, suggesting that the development of both new circuits and neuromodulatory systems is required for proper development of locomotor behaviors.

Although the parallel arrangement of differentially aged circuits that control distinct behaviors can explain how these behaviors can be controlled independently of one another, it does not explain how animals can coherently string together these behaviors. The coordination between early-born and late-born neurons has been described in a few neural systems. For example, through their widespread axonal arborizations, early-born 'pioneer' GABAergic neurons in the hippocampus exert a huge impact on the activity of neurons born afterwards, and in this way, these neurons play a crucial role in the synchronization of activity observed in the developing hippocampus (*Picardo et al., 2011*). Although these early-born neurons have been shown to exist in the adult hippocampus and to project heavily to the septum (*Villette et al., 2016*), it has yet to be revealed how their control over late-born neurons influences behavior in adulthood. The reticulospinal system in the zebrafish could serve as a good avenue for understanding the interactions that occur among age groups during behavior because it is composed of a relatively small number of identifiable neurons, many of which are known to be recruited during escape behavior (*Kimmel et al., 1982*; *Gahtan et al., 2002*). During escapes, fish transition from fast and crude whole-body movements to slower and refined tail-restricted movements (*Mirat et al., 2013*; *Marques et al., 2018*). The age-related recruitment pattern we observed in V2a descending neurons suggests that neural activity shifts from the early-born neurons contributing to the crude whole-body movements to the late-born reticulospinal neurons contributing to the refined tail-restricted movements. This raises the possibility that the recruitment of the early and late-born reticulospinal neurons are coordinated to ensure the smooth transition from the crude to refined locomotor patterns. Indeed, in adult goldfish and zebrafish, it's been shown that the activation of Mauthner cell, the earliest born reticulospinal neuron, influences

ongoing slow locomotor activity in the spinal cord (*Svoboda and Fetcho, 1996*; *Song et al., 2015*), further supporting the presence of such coordination. Thus, the reticulospinal system may permit detailed analyses of the interactions among age groups and their behavioral roles. In any case, detailed circuit analyses of early-born and late-born neurons in the context of behavioral development will be essential to understand how nervous systems organize the interactions between early-born and late-born circuits, and thus allow animals to produce coherent behaviors.

## Materials and methods

### Fish care
Zebrafish larvae were obtained from an in-house breeding colony of wild-type adults maintained at 28.5°C on a 14–10 hr light-dark cycle. Embryos were raised in a separate incubator but at the same temperature and on the same light-dark cycle. Embryos were staged (*Kimmel et al., 1995*) and only the ones with normal development were used in the study. All experiments presented in this study were conducted in accordance with the animal research guidelines from the National Institutes of Health and were approved by the Institutional Animal Care and Use Committee and Institutional Biosafety Committee of Janelia Research Campus (16-145).

### Transgenic fish
The following previously published transgenic lines were propagated to Casper background and used in this study (see *Supplementary file 1*): TgBAC(vsx2:GFP)(*Kimura et al., 2006*); TgBAC(vsx2:Kaede)(*Kimura et al., 2006*); TgBAC(vsx2:Gal4)(*Kimura et al., 2013*); Tg(UAS:Kaede) (*Davison et al., 2007*); Tg(UAS:synaptophysin-EGFP)(*Heap et al., 2013*); Tg(UAS:GCaMP6s) (*Muto et al., 2017*); Tg(mnx1:TagRFPT)(*Jao et al., 2012*); TgBAC(islet1:GFP)(*Higashijima et al., 2000*); Tg(Dbx1b:Cre, vglut2a:lRl-GFP)(*Koyama et al., 2011*). For examining cholinergic transmission in the Mauthner cell, we used the paralytic mutant, *relaxed*, to avoid the use of the cholinergic blocker α-bungarotoxin for the induction of paralysis (*Koyama et al., 2011*).

### Developmental imaging of hindbrain V2a neurons
TgBAC(vsx2:EGFP) was anesthetized in tricaine methanesulfonate (MilliporeSigma, E10521, St Louis, MO) dissolved in system water at 160 mg/L (hereafter referred as MS-222) and then embedded in 1.6% low melting point agar (MilliporeSigma, 2070-OP, MO). Volumetric images of hindbrain V2a neurons were acquired with a custom two-photon microscope. 25 × 1.1 NA objective lens (Nikon Instruments, CFI Apo LWD 25XW 1300 nm, NY) was used and EGFP was excited at 940 nm (MKS Instruments, Mai Tai HP, Deep See, MA). For each developmental time point, a separate group of fish was used to avoid the potential developmental effects of the imaging procedure (>6 fish per time point). For time lapse imaging, fish were kept in 0.5% agar and temperature was maintained at 28.5°C with a temperature controller (Luigs and Neumann, TC07, Germany).

### Birthdating of hindbrain V2a neurons using a photoconvertible fluorescent protein
Fish expressing the photoconvertible fluorescent protein, Kaede, in V2a neurons (TgBAC(vsx2:Kaede)) were photoconverted at 24, 36, 48, 60, 72 and 96 hr post fertilization (hpf) to identify V2a neurons that existed at a given developmental time point using a procedure similar to the one described previously (*Kimura et al., 2006*; *Caron et al., 2008*). Briefly, UV light was shone on around 10 fish within a drop of system water for 30–60 s using a stereo dissection microscope (Olympus, MVX10, PA) equipped with a DAPI filter (Chroma Technology, 49901, VT) and a halogen bulb (Excelitas Technologies, X-Cite 120PC Q, MA). Then, the fish were transferred back into the incubator and housed in a box that filtered out UV wavelengths from the light within the incubator. This prevented photoconversion of additional developing neurons by the ambient light in the incubator. The fish remained in the incubator until 120 hpf, at which point they were imaged with a confocal microscope (Zeiss, LSM710, Germany) with 20 × 1.0 NA objective lens (Zeiss, W Plan-Apochromat 20x/1.0, Germany) (>8 fish per time point). A group of fish converted at 24 and 36 hpf (six fish per time point) were injected with a dye in the spinal cord (*Liu and Fetcho, 1999*) at 4 days post-fertilization (dpf) to label reticulospinal neurons. A far red dye was used to avoid contaminating the red

channel used for imaging photoconverted Kaede (Alexa Fluor 680 dextran, MW: 10,000, Thermo Fisher Scientific, MA). These neurons are identifiable across animals and are named based on their rostrocaudal positions (*Figure 2C*, left) and dorsoventral positions (*Figure 2C*, right) (*Kimmel et al., 1982*; *Mendelson, 1986*). We used the same naming convention used previously except for a V2a reticulospinal neuron that was born by 24 hpf in rhombomere 4. This cell was the most dorsal cell within V2a reticulospinal neurons in rhombomere four but was slightly ventral to MiM1. Just as MiM1 refers to a single identifiable cell that is just ventromedial to the Mauthner cell (*Mendelson, 1986*), we decided to refer to this neuron as dorsal MiV1.

## Optical backfill of hindbrain V2a neurons

Fish expressing Kaede in V2a neurons through the Gal4-UAS system were photoconverted in the rostral spinal cord (muscle segment 5 to 7) at 36, 60,84 and 108 hpf to identify hindbrain V2a neurons that had spinal projections at a given developmental time point as described previously (*Kimura et al., 2013*). Briefly, fish were anesthetized in MS-222 and embedded ventral-side up in 0.5% low-melting point agar and then the rostral spinal cord (muscle segment 5 to 7) was illuminated for 2 min with 405 nm LED (Lumencor, Spectra X, OR) in an inverted microscope (Nikon Instruments, Eclipse Ti-E, NY) using a 60x objective lens (Nikon Instruments, CFI Plan Apo VC 60XWI, NY). This procedure was repeated three to five times with an interval of 15–20 min. The fish were then kept in the dark until 120 hpf and imaged with a confocal microscope (Zeiss, LSM 710, Germany) (>6 fish per time point). The expression of Kaede in the earliest born hindbrain V2a neurons, driven by the Gal4/UAS system was delayed roughly 6 hr relative to the one driven directly by the V2a promoter (data not shown). Theoretically, it is possible that some descending neurons are not labeled with this procedure due to 1) inefficient transport on small axons, or 2) fast turnover back to unconverted Kaede. However, it is unlikely that we are missing cells because of insufficient diffusion time (36 hr to 84 hr) because the converted Kaede diffused to the cell bodies of the caudal V2a neurons that are previously speculated to have thin axons in less than 12 hr (*Video 1*). Turnover time of converted Kaede was more than 96 hr in the early-born population (see *Figure 2—figure supplement 1* for example). This is longer than the diffusion time we used for our optical backfill experiments (36 hr to 84 hr). Thus, the second possibility is also unlikely.

## Two-photon calcium imaging of GCaMP6s fluorescence in hindbrain V2a neurons

The genetically encoded calcium indicator GCaMP6s (*Chen et al., 2013*) was expressed in hindbrain V2a neurons using the Gal4-UAS system (Tg(vsx2:Gal4; UAS:GCaMP6s)). Fish at 120–132 hpf were paralyzed with α-bungarotoxin (MilliporeSigma, 203980, MO) and then embedded in 1.6% low-melting point agar. The agar around the tail and the nostrils was released using a fine tungsten pin. Axial motor nerve activity was recorded through fire-polished glass pipettes placed on the dorsal inter-myotomal cleft on both sides. A gentle suction (~15 mm Hg) was applied through a pneumatic device to gently break the skin. The signal was amplified (1000x) and bandpass filtered (100–1000 Hz) through an extracellular amplifier (NPI, EXT-02B, Germany) and digitized at 6 kHz with PCIe-6363 (National Instruments, TX) using a custom C# program (Source Code File 1). Clear motor activity was typically observed within 20 min. The whole hindbrain was imaged with a custom two-photon microscope equipped with a resonant scanner (Thorlabs, MPM-2PKIT) and a piezo objective scanner (PI, P-725K129, Germany) at a volume rate of 2 Hz (512 × 256, 30 slices, 7 µm z step) using Scan-Image (Vidrio Technologies, VA). 940 nm, 80 MHz femtosecond laser pulses were used to excite GCaMP6s (MKS Instruments, Mai Tai HP, Deep See, MA). A gradual mechanical stimulus was delivered by gradually contacting the rostral part of the head with a fire polished glass pipette controlled by a motorized manipulator (Luigs and Neumann, Junior RE, Germany). A brief electrical stimulus (0.5–1 ms in duration, 2–10 V) was delivered through a glass electrode placed on the side of the head using a stimulus isolator (Digitimer Ltd., DS-2, England). The inter-stimulus interval was from 50 s to 2 min. The stimulus amplitude was adjusted to evoke strong motor activity consistently. Each experiment lasted between 20 and 40 min and thus contained 2400–4800 stacks.

## Analysis of two-photon calcium imaging data

Two-photon data was first corrected for mismatch between the odd and even scan lines by cross correlation if necessary and then the shift of the sample was corrected by cross correlation to a reference volume. Then GCaMP6s signal from identifiable V2a neurons were examined in relationship to fictive swim bouts. The regions of interest (ROI) for V2a-positive reticulospinal (RS) neurons (Mid2i, Mid3i, RoM2, RoM3, RoV3, dorsal MiV1, and ventral MiV1/2) were drawn based on their distinct segmental distribution, dorsoventral position and soma morphology. First, dorsal early-born V2a RS neurons were identified based on their large and laterally displaced cell bodies and used as a landmark for each rhombomere (RoM2, RoM3, dorsal MiV1 and Mid2i, Mid3i) (*Video 1*). Then, the younger ventral V2a RS neurons in each rhombomere were identified based on their smaller cell bodies ventral to the dorsal early-born neuron. This was confirmed further with spinal backfill (*Kimmel et al., 1982*). The ROIs for caudal hindbrain V2a neurons were drawn based on their stereotypical positions as examined in the birthdating analysis: the early-born caudal neurons were displaced laterally from the rest of caudal V2a neurons while the late-born caudal neurons were located dorsally, close to the midline. Then the fluorescence time course (F(t)) for each ROI was extracted and the baseline fluorescence ($F_0$) was estimated as the bottom 20th percentile of the whole timeseries. Then $\Delta F(t)/F_0$ was calculated as follows.

$$\frac{\Delta F(t)}{F_0} = \frac{F(t) - F_0}{F_0}$$

To detect weak spontaneous swimming activity reliably, the axial motor nerve activity was processed by taking a windowed standard deviation of the original signal with a 10 ms moving window as described previously (*Ahrens et al., 2012*). Then, a threshold for swimming activity was selected to detect the weak spontaneous swimming activity reliably from the transformed signal. In the gradual mechanical stimulus experiment, the latency of strong swimming activity from the onset of the stimulus was not consistent across trials. Thus, we used the following procedure to detect the strong bursting activity. For each swim bout, the maximum burst amplitude of the transformed signal was extracted independently for left and right channels (*Figure 3B* i). Then, the distribution of the maximum burst amplitude was log-transformed to reduce skew in the distribution and fitted as a Gaussian mixture distribution with two components to estimate the distribution of the maximum burst amplitude that corresponds to weak spontaneous swims. This estimated probability distribution corresponding to weak spontaneous swims was used to compute a value whose probability of falling in this distribution is less than 0.1% (*Figure 4B* i). The swim episodes with the maximum burst amplitude above this value were categorized as strong swims induced by the stimulus (*Figure 4B* i; *Figure 3C*, 'Push'). The detected episodes matched with the slow and strong bursting activity (as assessed by eye). In the electrical pulse stimulus experiment, fast swim episodes were elicited reliably by the stimulus and were categorized as shock-induced fast swims (*Figure 3C*, 'Shock'). The remaining spontaneous swim episodes were categorized as spontaneous swims (*Figure 3C*, 'Sponta swim'). To derive the $\Delta F/F_0$ response related to these swim events (*Figure 3C*), $\Delta F/F_0$ signal was aligned at the onsets of a given swim type and averaged over trials for each cell. Only the fictive motor signal ipsilateral to the cell being examined was considered for the following reasons: 1) hindbrain V2a neurons are primarily ipsilaterally projecting neurons (*Kinkhabwala et al., 2011*; *Cepeda-Nieto et al., 2005*). 2) most of the neurons related to the strong swims responded only when there is a strong axial motor activity on the ipsilateral side (*Figure 3A and B*). The peak of the mean $\Delta F/F_0$ response was detected for each neuron and pooled for a given cell type to test if they showed activity consistently during a given swim type (*Figure 3C*, top row). To test if swim types had effects on the activity of a given cell type, an one-way ANOVA with a factor for swim types was used. All possible comparisons were tested with Tukey's multiple comparison test (*Figure 3C*, bottom row).

To map the hindbrain V2a neurons recruited during these swim bouts in an unbiased way, voxel-level regression analysis was done with regressors representing the distinct types of swim bouts. Each regressor was constructed by convolving the corresponding fictive motor signal (as defined above) with a GCaMP6s impulse response function modeled as the rise and decay exponentials (0.5 s rise and 2 s decay) and then standardized for a given experimental session by subtracting its mean from its values and then dividing these new values by its standard deviation (*Figure 4A* ii, 4B ii).

Voxel time course (Y) was fitted with these standardized regressors (X) using the following linear model.

$$Y = X\beta + \varepsilon$$

The standardized coefficient (β) was estimated by ordinary least square and then $T$ value for each voxel was calculated based on standardized coefficient (β) and residual (ε). Correction for multiple comparisons was done with the false discovery rate (FDR) (*Miri et al., 2011b*). The threshold for $T$ maps was set at $P_{FDR} < 0.05$.

## Registration of volumetric images to Zebrafish Brain Browser atlas

Maps of birthdate, descending projection and activity of hindbrain V2a neurons were registered to Zebrafish Brain Browser (ZBB) atlas (*Marquart et al., 2015*; *Marquart et al., 2017*) using the procedure described in *Marquart et al. (2017)* with minor modifications. Briefly, the volume of vsx2:Gal4 provided by ZBB was used as a reference for all the registration of V2a population. Then the reference images for each map was created using either the summed volume of unconverted and converted Kaede for the birthdate and descending projection maps, or the time-averaged volume for activity maps. Affine and symmetric diffeomorphic registrations were performed using Advanced Normalization Tools (*Avants et al., 2008*; *Avants et al., 2011*) using the parameters described for live samples (*Marquart et al., 2017*).

## Spinal imaging of descending fibers from hindbrain V2a neurons

The photoconvertible protein, Kaede, and GFP fused to synaptophysin were expressed in V2a neurons using the Gal4/UAS system (Tg(vsx2:Gal4;UAS:Kaede;UAS:synaptophysin-EGFP)). Hindbrain V2a neurons were photoconverted with the same set up used for optical backfill but with a shorter total exposure time (1 round of 1 min exposure). The photoconversions were done at 42 and 96 hpf to visualize the spinal projections of hindbrain V2a neurons that existed at a given developmental time point. The fish were then kept in the box that filtered out UV until the imaging at 120 hpf. A series of confocal images were taken in the hindbrain and the spinal cord from the dorsal side (Zeiss, LSM710, Germany). In a separate set of experiments, hindbrain V2a neurons were photoconverted at 42 hpf and spinal motoneurons in the rostral spinal cord were labeled by the injection of a far-red dye (Thermo Fisher Scientific, Alexa Fluor 680 dextran, MW: 10,000, MA) in the axial muscles at 108 hpf. In yet another set of experiments, spinal interneurons in the rostral spinal cord were labeled instead by the injection of the far red dye in the caudal spinal cord (28th to 30th muscle segment) in a manner described previously (*Hale et al., 2001*).

## Analysis of distribution of converted Kaede in the hindbrain and spinal neuropil

Volumetric images of V2a neurons labeled with converted and unconverted Kaede in the hindbrain and the spinal cord were 3D-rendered in Imaris (Bitplane, Switzerland). Then maximum intensity projection views from the dorsal side were created in the area of interest. The views were carefully oriented based on anatomical landmarks such as axons from V2a neurons to minimize the error from tilt of the 3D volume. Then the intensity distribution in the neuropil along the mediolateral axis in the projection images was calculated in Fiji (Fiji is just Image J) from a line profile extending from the lateral surface of the cell bodies of the most laterally located V2a neurons to the lateral surface of the neuropil from V2a neurons. The line thickness was set to 20 µm in Fiji to average the intensity profiles over 20 µm in the rostrocaudal direction. The intensity of converted Kaede and the mediolateral position were normalized in each fish and the average intensity profiles and their standard errors were plotted as a function of the normalized mediolateral position in the neuropil.

## Hindbrain whole-cell recordings

Patch-clamp recordings of hindbrain neurons were performed with a procedure similar to the one described previously (*Kimura et al., 2013*), but with some modifications. Five-day-old fish were paralyzed with α-bungarotoxin (MilliporeSigma, 203980, MO) that was dissolved in system water (1 mg/ml). After successful paralysis, larvae were anesthetized with MS-222 and then secured with etched tungsten wires (pins) through the notochord to a Sylgard-coated glass-bottom dish containing

extracellular solution (134 mM NaCl, 2.9 mM KCl, 1.2 mM MgCl$_2$, 2.1 mM CaCl$_2$, 10 mM HEPES, and 10 mM glucose, adjusted to pH 7.8 with NaOH). Then, the head was rotated and secured ventral side up with tungsten pins placed through the ears and the rostral part of the jaw (*Figure 7A* ii). The ventral surface of the hindbrain was carefully exposed by removing the notochord using an tungsten pin and fine forceps. The skin of the middle region of the body was removed with a pair of forceps to gain access to the peripheral nerves from spinal motoneurons. Locations of motor nerve recordings were from the 10th to 15th muscle segment.

Neurons were targeted based on fluorescence and scanned Dodt gradient contrast images acquired with a custom two-photon microscope. 40x (Nikon Instruments, CFI Apo 40XW NIR, NY) or 25x (Leica, HCX IRAPO L 25x/0.95 W, Germany) objective lens was used. Descending hindbrain neurons were identified by the fluorescence signal from Tg(vsx2:EGFP), Tg(vsx2:Kaede) or spinal backfill (see above). MiV1 neurons were located in rhomobomere 4, medial and ventral to Mauthner cells which can be easily identified by Dodt gradient contrast. The subclass of MiV1 neurons was targeted based on photoconversion of Kaede (see above) and its stereotypical position. We also quantified dorso-ventral position based on volumetric images acquired after the recording as a proxy of birthdate, taking advantage of the fine-scale birthdate-related topographical organization. First, we made a coronal optical section including all the neurons in this cluster. The plane of the section was carefully adjusted so that it was parallel to the V2a processes running lateral to MiV1 neurons. Then, we identified dorsal MiV1 based on its close proximity to M-cell, its soma larger than the nearby younger cells, and the presence of photoconverted Kaede. We then defined normalized dorso-ventral position with '0' being the dorsal edge of dorsal MiV1 and '1' being the ventral edge of the most ventral MiV1. The cells other than dorsal MiV1 with the position smaller than 0.5 were categorized as middle MiV1, while those with the position larger than 0.5 were categorized as ventral MiV1.

Whole-cell recordings were established using the standard procedure and the signals were recorded using EPC 10 Quadro amplifier (HEKA Electronik, Germany) and PatchMaster (HEKA Electronik). The resistance of the electrode was 8 to 12 MOhm. The intracellular solution contained (in mM) 125 mM K-gluconate, 2.5 mM MgCl$_2$, 10 mM EGTA, 10 mM HEPES and 4 mM Na$^2$ATP adjusted to pH 7.3 with KOH. A junction potential using this intracellular and extracellular solution (see above) has been calculated at 16 mV, which would result in a shift of measured potentials 16 mV in the negative direction. Because this would not affect our conclusions, we did not correct for it. The solution also contained 0.01% of Alexa Fluor 568 hydrazide or Alexa Fluor 647 hydrazide (Thermo Fisher Scientific). Z stacks were acquired with a custom two-photon microscope to confirm morphology and locations of the recorded neurons. Input resistances were calculated from an average of 5 hyperpolarizing square current pulses between 20 and 80 pA. This range of current injection produces a linear current-voltage response in all the MiV1 neurons examined.

Electrophysiological analysis of recruitment of MiV1 neurons was performed in a manner similar to the previous studies (*McLean et al., 2007*; *McLean et al., 2008*). Fast fictive swimming was induced by a brief electrical stimulus (<1 ms in duration at 1–10 V) delivered via a pair of tungsten electrodes (A-M systems, WA) placed near the tail. Slow swimming often occurs spontaneously but is also induced by flashes of light. Cycle frequencies of axial motor activity were computed by taking the reciprocals of the time interval between each burst and the next one. The spiking and subthreshold activity that precede each cycle were extracted and their relationship to cycle frequency was examined as follows. Spikes were detected based on the threshold determined for each cell. Subthreshold activity was examined based on the lowpass-filtered (60 Hz) voltage trace, which effectively isolated rhythmic subthreshold activity during swimming. The maximum depolarization for each cycle was calculated by subtracting its peak from the baseline which was defined as the minimum voltage in the 200 ms time window before the swim onset. In order to statistically test if each age group changes spiking and subthreshold activity as a function of cycle frequency, each cycle was classified as fast (>35 Hz) or slow (<35 Hz) cycle and fitted with the following linear mixed models with a random effect for cell using the *nlme* package in R (https://cran.r-project.org/web/packages/nlme/index.html). Number of spikes was log-transformed to satisfy the assumptions of the linear model. The effect of cycle speed was tested by comparing the activity for each age group with correction for multiple comparisons using the *multcomp* package in R (https://cran.r-project.org/web/packages/multcomp/index.html).

$$log\,(number\,of\,spikes) \sim cycle\,speed * age$$

$$membrane\,depolarization \sim cycle\,speed * age$$

## Single-cell analysis of spinal projection

Electroporation of Alexa Fluor 647 dextran dye (MW 10,000; Thermo Fisher Scientific) into single neurons was performed as described previously (*Koyama et al., 2011*). MiV1 neurons were targeted based on GFP expression in Tg(vsx2:GFP) and/or spinal backfill with Texas Red dextran dye (MW 10,000; Thermo Fisher Scientific) along with scanned Dodt gradient contrast under a custom two-photon microscope at 5 days old. One day after electroporation, the hindbrain was imaged with a confocal microscope (Zeiss, LSM 710) from the dorsal side to confirm the dorsoventral position of the electroporated MiV1 neurons. Multiple z stacks from the side were also imaged from hindbrain to the caudal end of spinal cord to cover the entire spinal projection. Stacks were then stitched together with stitching plugin available through Fiji (*Preibisch et al., 2009*). When spinal backfill was used to visualize descending neurons, the spinal segments caudal to the injection site (21st to 23rd muscle segment) were excluded from imaging. Stitched stacks were 3D rendered in Imaris (Bitplane, Switzerland). To visualize the spinal projection in coronal view, axonal arborization was reconstructed with neuTube (http://www.neutracing.com/) and then visualized in Imaris. Three to four muscle segments were used to render coronal views in the rostral spinal cord (6th to 9th muscle segment) and in the middle spinal cord (14th to 17th muscle segment). All the cells examined in the study had their main axon in the ventral spinal cord and had collaterals innervating the dorsal spinal cord. The dorsal extent of the axonal arborization was quantified relative to the total dorsoventral extent of spinal cord as described previously (*McLean et al., 2007*). To statistically examine if the extent of spinal arborization depends on hindbrain cell type and the position in the spinal cord, the dorsal extent of arborization was fitted using the following linear mixed effects model with a random effect for cell using *nlme* package in R.

$$Dorsal\,extent\,of\,axon\,collaterals \sim hind\,brain\,neuron\,types * position\,in\,the\,spinal\,cord + (1|Cell\,ID)$$

Putative presynaptic terminals were identified based on their characteristic puncta-like structure. Then their apposition to fluorescently labeled spinal neurons was examined. Primary motoneurons were identified using Tg(mnx1:TagRFPT). Secondary motoneurons were identified using TgBAC (islet1:EGFP). Spinal V2a neurons (also called circumferential ipsilateral descending neuron (CiD)) were identified using TgBAC(vsx2:EGFP). Multipolar commissural neurons (MCoD) were identified by spinal backfill from the caudal spinal cord (see above). To statistically examine if hindbrain cell type explained the variance in their spinal innervation patterns, the presence of innervation to the spinal neurons was fitted with the following generalized linear mixed effects model with a random effect for cell and a logit link function for a binomial distribution using *lme4* package in R (https://cran.r-project.org/web/packages/lme4/index.html).

$$Presence\,of\,innervation \sim hindbrain\,neuron\,types * spinal\,neuron\,types + (1|Cell\,ID)$$

## Paired whole-cell recordings of hindbrain and spinal cord neurons

Whole-cell recordings from hindbrain neurons were done as described above. The spinal cord was also exposed by removing the muscles overlying the cell of interest using the standard procedure described previously (*Drapeau et al., 1999*). Dorsal and ventral MiV1 neurons were targeted as described above. Mid3i was identified based on the GFP signal from Tg(vsx2:EGFP), its stereotypical segmental location (rhombomere 6) and morphology (laterally elongated soma and its position relative to nearby reticulospinal neurons in the same segment). Mauthner cell was identified based on its segmental position (rhombomere 4) and its large soma which is clearly visible under Dodt gradient contrast. Spinal motoneurons and interneurons (MCoDs) were targeted with Dodt gradient contrast and/or fluorescence either from backfill (see above) or from transgenic lines (Tg(mnx1:TagRPFT) for motoneurons; Tg(Dbx1b:Cre, vglut2a:lRl-GFP) for MCoDs). Primary motoneurons are dorsally located large cells close to the lateral surface of the spinal cord. MCoDs are ventrally located smaller cells but also close to the lateral surface of the spinal cord. PMNs were sampled at muscle segment 17 to 19. MCoDs were sampled at muscle segments 11 to 13. To examine if a given connection is monosynaptic, extracellular solution with a high concentration of divalent cations (4x $Mg^{2+}$ and 4x

$Ca^{2+}$) was perfused to increase the firing threshold of potential relay neurons. Only the connection between Mauthner cell and MCoD was almost completely abolished, which is also consistent with its long latency. Thus, this connection was excluded from subsequent analysis. To examine the composition of electrical and chemical synapses for a given connection, chemical synapses were blocked by a cocktail of D-AP5 (Tocris, 0106, MN) and NBQX (Tocris, 0373, MN) for glutamatergic synapses and by mecamylamine (Tocris, 2843, MN) for cholinergic synapses.

To statistically examine if hindbrain and spinal cell types had significant effects on the presence of monosynaptic connections, the presence of monosynaptic connection was fitted with the following generalized linear model with a logit link function for a binomial distribution using *lme4* package in R.

$$Presence\ of\ monosynaptic\ connection \sim hindbrain\ neuron\ types * spinal\ neuron\ types$$

The differences in conduction velocity, postsynaptic potential (PSP) amplitude and PSP half decay time across connected pairs of the monosynaptic responses were statistically examined with Kruskal-Wallis test. Post-hoc comparisons were done with Dunn's multiple comparison test.

## Targeted two-photon laser ablation

We ablated two groups of V2a neurons based on the developmental and recruitment analyses we performed (*Figures 2–4*). The first group consisted of the early-born V2a neurons that were recruited during strong swims. Procedurally, they were defined as hindbrain neurons that contained photoconverted Kaede at 60 hpf after photoconversion at 24 hpf in Tg[vsx2:Kaede] (*Figure 10B* i). The second group consisted of the late-born V2a neurons in the caudal hindbrain that were recruited during spontaneous weak swims. Procedurally, they were defined as neurons that did not contain photoconverted Kaede at 60 hpf after photoconversion at 42 hpf and met the following spatial criteria (*Figure 10B* iv). First, their somata were caudal to rhombomere six where the early-born neurons were located in dorsal hindbrain (*Kinkhabwala et al., 2011*) Second, they were dorsal to the early-born neurons in the caudal hindbrain. These neurons born after 42 hpf probably do not include all the caudal neurons active during weak swims (see *Figure 5A* vi and *Figure 2F*). Thus, we may underestimate the contribution of the caudal V2a neurons to spontaneous weak swims. However, we chose this to make sure there is no unintended damage to the most lateral early-born neurons so that we can see distinct locomotion phenotypes for each ablation group. The time of ablation was matched in the two groups to minimize the difference in the procedure.

For every fish that was randomly selected for ablation from the pool of fish that had been photoconverted, one or two more fish were chosen in the same way to serve as a negative control(s). The fish to be ablated was anesthetized in MS-222 for 2 min before being mounted dorsal-side up in 1.6% low melting point agarose and placed under our custom two-photon microscope where the ablations took place. Both the fish in which the ablations were carried out and the fish chosen as control were kept under anesthesia for the entire duration of the ablations. Before proceeding to ablations, we first imaged hindbrain V2a neurons labeled with unconverted and converted Kaede using 1020 nm, 80 MHz femtosecond laser pulses (MKS Instruments, Insight, Deep See, MA) with a 25x objective lens (Leica, HCX IRAPO L 25x/0.95 W, Germany). Then, we proceeded to ablations starting from the most ventral target cells to increasingly more dorsal cells to minimize light scattering that could result from preceding ablations. For a given depth, each target cell was exposed to femtosecond laser pulses for 5 ms with spiral scans centered on its soma (outer diameter, 3 um) before targeting the next target after a minimum 5 s interval using ScanImage (Vidrio Technologies, VA). We used 920 nm or 1020 nm laser (196 mW or 142 mW after objective lens, respectively). This procedure was repeated up to ten times until the cells at a given depth were deemed ablated based on visual inspection. After ablating all the cells of interest, we imaged hindbrain V2a neurons again to assess the quality of ablation. Then, the fish was gently removed from the embedding agarose and both this fish and the control fish were housed singly in separate dishes until later use on the day of behavior imaging, which was at 5 dpf. To examine the potential recovery of target cells after the ablation, a subset of the ablated fish and the control fish was selected for imaging at four dpf. From the time of photoconversion till the day of behavior, the dishes containing the fish were enclosed in our UV-opaque box to prevent further conversion of Kaede.

## Behavioral assays

At 5 dpf, ablated and control fish were transferred singly and in random sequence to a 50 mm diameter circular arena where they were allowed to swim freely while being imaged from above with a high-speed camera (Miktrotron, MC-1362, Germany). To confine the movements of the fish to the yaw (x-y) plane, the depth of the water in the arena was limited to 2 mm. Before the start of imaging, each fish was allowed to acclimate to the arena for a minimum of 20 min before their behavior was imaged. If even after 20 min the fish spent a majority of the time being stationary or swimming very close to the edge of the arena (as they tend to do soon after being transferred into the arena from their home dish), then the wait period was increased until the fish started showing frequent swims in trajectories that took them far from the arena edge. Following this acclimation period, fish were first imaged at 300 frames per second (fps) for a total 5 min while they swam spontaneously. After this 5-min period, fish were imaged continuously while they were presented with a series of alternating vibration and dark flash stimuli - although in this paper we did not use data from dark flash responses - at 1 min intervals. A time period of 100 ms before and 1400 ms after each stimulus were imaged at 500 fps, whereas the intervening 1 min periods between stimuli were imaged at 30 fps. The vibration stimulus was delivered via a sound speaker in contact with the acrylic base plate upon which the fish arena was placed. The sound stimulus consisted of two cycles of 500 Hz sine wave at (peak-to-peak acceleration $\approx 100 \ ms^{-2}$). For the dark flash stimulus, the ambient light was turned off for 300 ms.

Images collected during the behavior sessions were first sorted by frame rate. For images collected at 30 fps we only tracked the fish's head position, whereas for images collected at 300- and 500 fps, we estimated the fish's head orientation and body curvature as well.

## Behavior tracking

To extract the fish's head position, we first computed a background image and subtracted this from all images to remove background from them. The background image was computed by averaging a set of 1000 temporally uniformly spaced images from the set of all images. After background subtraction, for convenience, the pixel values were multiplied by -1 so that the fish's eyes would go from being the darkest to the brightest spots in the image. The images were then smoothed by convolving each image with a 1 mm wide 2D Gaussian kernel. The convolved images were then segmented using an automatically estimated threshold (multithresh function in MATLAB) to isolate within each image a blob (contiguous set of pixels) of the brightest pixels that included the fish's eyes as well as the swim bladder. The centroid of these isolated pixels was used as the fish's head position. The centroid obtained in this manner was reliably located between the eyes and the swim bladder in all image frames and for frames in which the fish did not move, the centroid typically shifted by less than 0.17 mm ($\approx$ 3 pixels) between frames (confidence interval of 95%).

For quantifying the fish's tail curvature, we needed to reliably isolate as many pixels on the fish's body as possible so that the movements of the tail could be accurately tracked during swimming. For this, we used a custom MATLAB script (available upon request) that used an iterative procedure to obtain a threshold for image binarization and isolation of fish pixels.

Following the detection of fish pixels, the images were binarized by setting fish pixels to 1 and the rest of the pixels to 0. The MATLAB function bwmorph was then used on the binary images to 'thin' the fish pixels down to a set of connected pixels that spanned the length of the fish and coarsely bisected it into two lateral halves. To transform these 'midline' pixels into a smoother curve that better tracked the spine of the fish, we followed the procedure described in (*Huang et al., 2013*), wherein the integer-valued image coordinates of the midline pixels were weighted by the intensities of the surrounding pixels to obtain a new decimal-valued set of pixels. The curve obtained by this procedure was then interpolated using cubic splines to obtain a final smooth curve ($C$) of 50 points.

## Quantification of free swimming behaviors

To characterize swimming, we looked at changes in the body posture and head orientation over time. We used the midline curve ($C$) obtained using the procedure described above to characterize body posture. First, we computed tangent vectors $\hat{t}$ along the points on $C$ and used the angle differences between adjacent tangent vectors to compute local curvatures. We then cumulatively summed

these curvatures along the tail, starting from the point closest to the head centroid. This way we obtained a set of angles along the midline that represented the angular difference between the tangent at a given point along the midline and the tangent at head centroid. The total curvature ($K$) was computed as the sum of all the local curvature along the tail.

To obtain the head orientation, the first 10 points (the 10 points closest to the head centroid) of the midline curve $C$ were sampled and a straight line ($L$) was fit to those points. The head orientation ($\phi_H$) was then estimated as the absolute angle of the vector extending from the point on $L$ farthest from the head centroid to the point on $L$ closest to the head centroid.

## Identification of swim episodes

To identify distinct swim episodes, we used the total curvature time series, $K(t)$, after low pass filtering it at 70 Hz to remove high-frequency noise. We then used a custom MATLAB script to identify the times corresponding to the onset and offset of each swim episode. Briefly, the absolute function of the filtered curvature timeseries ($|K(t)|$) was convolved with a 200 ms gaussian kernel and in the first step the onsets of swim episodes were identified as the points where the transformed timeseries ($S(t)$) crossed and stayed above a threshold of 4o for at least 50 ms. Then, for each onset a corresponding offset was identified as the first point in $S(t)$ after this onset where the signal returns and stays below the threshold for at least 50 ms. Then, in a second step to improve on the loss of temporal resolution resulting from the convolution of $K(t)$ to get $S(t)$, we updated each onset by replacing it with the time of the peak in the second derivative of $S(t)$ that occurred immediately before this onset.

## Sorting swim episodes into distinct categories

To sort spontaneous swim episodes into distinct subtypes we used the amplitudes of the first bend within each swim episode because it was shown in a previous study (*Burgess and Granato, 2007*) that this single metric could be used to effectively categorize spontaneous swims into at least two distinct groups. To obtain the decision boundaries for sorting spontaneous swimming episodes from our dataset into distinct swim categories we extracted the first bend amplitude from each episode and fit a Gaussian Mixture (GM) model to the data using the scripts available in the scikit-learn library (http://scikit-learn.org/stable) in Python. For training the GM model, we used data from 15 fish (6056 swim episodes) in which V2a neurons had not been photoconverted (*Figure 10A*, iv-v) because photoconversion could have potentially altered swimming characteristics. Our model incorporated three Gaussian components because this resulted in the lowest value for the Akaike Information Criterion (AIC) (*Figure 10A*, iv bottom), indicating that our dataset consisted of three distinct spontaneous swim types. Finally, we used the trained GM model on the entire spontaneous swimming dataset to assign each episode to one of the three categories. To visually inspect the swim episodes from each category we used a custom script to temporally align all the episodes (curvature time series) within a category in such a way as to produce the highest correlations among them, and then plotted the traces atop of each other (*Figure 10A,vi*). For aligning escape episodes which have highly nonstationary beat frequencies, correlations were computed using a restricted time window of 80 ms after the episode onset. As with the training of the GM model, we only used data from non-photoconverted fish to make these plots. The plots showed that the swim episodes within each category were quite similar not just with respect to the amplitudes of their first bends, which we had used to predict swim categories, but with respect to subsequent bends as well. Furthermore, the overlaid traces revealed that swim episodes within a category were very similar to each other, but swim episodes across categories were quite distinct. This confirmed that as in the study by *Burgess and Granato (2007)*, we could use the amplitude of just the first bend of each swim episode to effectively categorize spontaneous swims into distinct subtypes.

In the case of escape swims, we treated them as belonging to a single category because they were all produced in response to the same vibration stimulus and because they are very rarely, if ever, observed in the absence of a stimulus. Thus, for comparing the individual bend amplitudes and periods between control and ablated fish, we did not sort escapes in the same way as spontaneous swims.

## Computing swim parameters from swim episodes

To evaluate the effects of ablations on swimming, we computed two types of swim parameters from each swim episode and compared these across ablated and control groups of fish. The first type of parameters were global in that each of these parameters described some aspect of a swim episode using a single number. In this study, the global swim parameters we examined were total swim distance per episode, episode duration, mean swim velocity per episode, maximum swim velocity per episode, total number of body bends per episode, and onset latency of the episode (only for escapes because these were triggered using a stimulus of very short duration). The other type of swim parameters we computed were local in that they were a set of numbers that described a swim episode at different stages. Here, the local parameters we examined are the amplitudes and durations (or periods) of the different body bends that comprise a swim episode.

To estimate the traveling distance per episode, we tracked the spatial position of a point on the fish's body just caudal to the swim bladder. We chose to track this point instead of the head centroid because it captured the fish's translation in space without being susceptible to oscillations of the head, such as those conspicuously observed during escapes. The total distance per episode was then computed as the sum of the distances traveled by the tracked point from one frame to another over all the image frames containing a single swim episode. The duration of an episode was simply the time elapsed from the detected onset to the offset of the episode (see above), and the mean swim velocity was obtained by dividing the total swim distance per episode by the duration of the episode. To compute the maximum swim velocity, the temporal derivative of the fish's spatial position was first computed to obtain the instantaneous swim velocity. The maximum of this timeseries during the course of an episode was the maximum swim velocity per episode. The number of bends per episode was computed by detecting and counting the total number of peaks in the curvature timeseries corresponding to a single swim episode.

Bend amplitudes and bend periods within a swim episode were computed after detecting peaks within the curvature timeseries as follows:

$$A(n) = |K(t_n) - K(t_{on})| \quad when \; n = 1$$

$$A(n) = |K(t_{n+1}) - K(t_n)| \quad when \, n > 1$$

$$P(n) = t_n - t_{on} \; when \; n = 1$$

$$P(n) = t_{n+1} - t_n \; when \, n > 1$$

where $n$ corresponds to bend number, $A(n)$ and $P(n)$ are the $n^{th}$ bend amplitude and period respectively, $t_n$ is the time at the $n^{th}$ peak, and $t_{on}$ is the time of swim onset. Peak angular velocities within a swim episode were computed in a similar manner from the angular velocity time series $\omega(t)$, which is the temporal derivative of the timeseries $K(t)$.

## ROC analysis of swim episode stereotypy

To quantitatively examine how the stereotypy of swim episodes within each swim category changes over the course of an episode, which was one of the criteria used to limit the analysis of bend amplitude and periods first 10 bends of an episode (see Results), we utilized ROC (Receiver Operating Characteristic) analysis. We first computed the probability distributions of the peak times of each of the bends of a swim episode (*Figure 10—figure supplement 3A*) for all swim categories. Here, the peak time of a bend refers to the time elapsed from the onset of the episode to the time when that bend reaches peak amplitude. Then, from these probability distributions, we estimated the discriminability of pairs of adjacent bends using ROC analysis. For each pair of adjacent bends up to the 10th bend, we used the probability distributions of their peak times to compute ROC curves (*Figure 10—figure supplement 3B* i). The area under the curve (AUC) for an ROC curve generated for a given bend pair (say, the second and third bend) indicated how discriminable that pair of bends was with respect to their peak times. To examine how discriminability of pairs of bends changes for each pair of successive bends, we plotted the AUC values computed for all the bend pairs against bend pairs ordered in succession (*Figure 10—figure supplement 3B* ii).

## Statistical assessment of the effects of ablations

To determine if the global and local swim parameters we computed were significantly different in control and ablated fish, we carried out statistical tests using software packages implemented in R. The data was first fit using a mixed effects model (nlme package) in which the swim parameter of interest (swim distance, bend amplitude, bend period, etc) was treated as the response variable while the categorical variables treatment, ablation group, and swim type were treated as the explanatory variables. The variable treatment took two values that indicated if the source of a given data sample was an ablated or a control fish. The variable ablation group also assumed two values that indicated if a data point had been collected from a fish in which old (birth-time < 24 hpf) or young (42 hpf <birth time<60 hpf) V2a neurons had been ablated. Finally, the variable swim type indicated which of the aforementioned swim categories the data point belonged to. In addition to the above explanatory variables, the identity of the fish was modeled as a random variable; the values within this last variable uniquely identified each fish that contributed to our dataset. The model also took into account interactions between the independent variables. We used this model-based approach to separately assess the effects of ablation on any of the variables of interest to us. This approach requires that the variables of interest be normally distributed, so to check for normality, we generated quantile-quantile (Q-Q) plots of the data, fit a straight line to the points and computed the $R^2$ value to assess how good the fit was. $R^2$ values greater than 0.8 were deemed good. Based on this criteria, all the swim parameters we looked at except escape onset latencies followed a normal distribution. However, when latencies were log-transformed and normality assessed on the transformed data, the $R^2$ value exceeded our cutoff value of 0.8. So, the statistical test was carried out on log-transformed escape latencies. Significance was assessed at $p<0.01$, and the Holm correction for multiple comparisons was made.

$$swim\,parameter \sim treatment * ablation\,group + swim\,type + (1/fishID)$$

Here, swim parameter can be any of the following: total swim distance per episode, episode duration, mean swim velocity per episode, maximum swim velocity per episode, total number of bends per episode, onset latency of escape, bend amplitude, bend period.

## Acknowledgements

We are grateful to the present and past members of our lab, especially to Jamien Shea who supported the early phase of this work and Masashi Tanimoto who provided critical feedback to the manuscript. We are also thankful to James Fitzgerald, Tzumin Lee, Misha Ahrens and Erik Snapp for providing critical feedback and to the members of Misha Ahrens' lab for their feedback during joint lab meetings. We thank all the members of the aquarium facility at Janelia for providing excellent care to the fish used in this study. We greatly appreciate the expertise and equipment provided by Janelia Experimental Technologies, especially by Vasily Goncharov and Christopher McRaven. Finally, we thank the scientific and administrative community at Janelia for creating an environment that allows for the focused pursuit of fundamental scientific questions.

## Additional information

### Funding

| Funder | Author |
| --- | --- |
| Howard Hughes Medical Institute | Avinash Pujala Minoru Koyama |

The funders had no role in study design, data collection and interpretation, or the decision to submit the work for publication.

### Author contributions

Avinash Pujala, Data curation, Software, Formal analysis, Validation, Investigation, Visualization, Methodology, Writing—review and editing; Minoru Koyama, Conceptualization, Resources, Data

curation, Software, Formal analysis, Supervision, Funding acquisition, Validation, Investigation, Visualization, Methodology, Writing—original draft, Project administration, Writing—review and editing

### Author ORCIDs
Avinash Pujala http://orcid.org/0000-0001-8758-1634
Minoru Koyama http://orcid.org/0000-0002-9774-9223

### Ethics
Animal experimentation: All experiments presented in this study were conducted in accordance with the animal research guidelines from the National Institutes of Health and were approved by the Institutional Animal Care and Use Committee and Institutional Biosafety Committee of Janelia Research Campus. (16-145).

### Decision letter and Author response
Decision letter https://doi.org/10.7554/eLife.42135.041
Author response https://doi.org/10.7554/eLife.42135.042

## Additional files

### Supplementary files
• Source code 1. C# program for delivering stimulus and monitoring fictive motor behaviors.
DOI: https://doi.org/10.7554/eLife.42135.037

• Supplementary file 1. Transgenic lines used in this study.
DOI: https://doi.org/10.7554/eLife.42135.038

• Transparent reporting form
DOI: https://doi.org/10.7554/eLife.42135.039

### Data availability
Source data files have been provided for numerical data that are represented as graphs in Figure 3C, 7D, 7E, 7F, 8E, 9J, 9K, 9L, and 10C.

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
