## [Decision Letter]

Thank you for submitting your article "Chronology-based architecture of descending circuits that underlie the development of locomotor repertoire after birth" for consideration by *eLife*. Your article has been reviewed by three peer reviewers, including Vatsala Thirumalai as the Reviewing Editor and Reviewer #1, and the evaluation has been overseen by Ronald Calabrese as the Senior Editor. The following individuals involved in review of your submission have agreed to reveal their identity: Vatsala Thirumalai and Claire Wyart.

The reviewers have discussed the reviews with one another and the Reviewing Editor has drafted this decision to help you prepare a revised submission.

Summary:

The manuscript from Pujala and Koyama is an impressive investigation of the development of descending motor control focusing on the V2a neuronal population in the hindbrain. The studies demonstrate that birth order of spinally projecting hindbrain V2a neurons causes anatomical layering of neuronal synaptic connections, and thus differentiates the role of early- versus late-born neurons in more primitive versus sophisticated behaviors. The authors combine a description with great precision of the developmental origin and organization of V2a interneurons in the hindbrain, together with a functional investigation relying on calcium imaging, functional anatomy and electrophysiology to argue that chronology based wiring could be a general principle by which circuits are wired resulting in gradual acquisition of behaviors.

Essential revisions:

The reviewers were unanimous in their praise for the meticulous and thorough investigation undertaken by the authors. However, they also raised a number of concerns that have to be addressed satisfactorily before the paper can be accepted.

1) An analysis of hindbrain V2a recruitment with respect to locomotor speeds during escapes, spontaneous swims and struggles. The manuscript covers many aspects of the organization, anatomy and function of V2a hindbrain interneurons without referring ever to locomotor speed or tail beat frequency. Work from the Fetcho, El Manira and McLean labs have previously shown spinal V2a interneurons to be recruited as function of speed. Is there a similar recruitment at the hindbrain level? Further, escapes in the larva consist of a C bend followed by fast swimming that largely transitions to slow swimming (Mirat et al., 2013; Knafo et al., 2017; Marques et al., 2018). This implies that during the "crude" escape, locomotion starts fast and transitions to slow. How is this transition implemented at the level of the hindbrain V2a interneurons?

2) Registration of images against a standard zebrafish brain atlas. The recent efforts made by the community to generate a brain atlas for zebrafish larva (see for example the ZBB from Harry Burgess) is critical here for registering the position of the soma and neuropile of V2a interneurons. In many instances (such as the caudal medial V2a cells), it is not clear where in the dorsoventral axis the soma of V2a cells are located. When presenting specific groups of V2a interneurons, the authors need to register their stacks in the ZBB atlas (or equivalent brain atlas) and add supplemental videos showing where the different V2a types referred to in the manuscript are located. This will also be helpful in Figure 4, where it is not clear how cells were classified as rostral, caudal, dorsal or ventral. It is also not clear how the authors ensured that they recorded from the same cells across fish. In addition, for Figure 7 and Figure 8, authors should analyze dorso-ventral position of the spinal targets as a further indicator of whether there is speed-dependent recruitment of the MiV1 cluster.

3) Analysis of behaviors after ablation: The authors analyzed only the initial 6 bends of each swim episode? Why not all of them? Also, what is the result when you look at more global parameters such as total distance traveled and speed for both escape and the 3 spontaneous swim types? Furthermore, did you observe a difference in latency for the escape response? Did you analyze struggles after the ablations?

4) Rewriting the Discussion section to include a more comprehensive view of circuit development and emergence of behaviors. There are several issues that have not been adequately addressed by the authors in the Discussion section:

- The existence of dedicated interneuronal and motoneuronal classes for mediating escapes is well established in zebrafish through work from multiple labs. Could it be that, escape, being critical for survival matures very early and has dedicated pathways and that the correlation with chronology is incidental? In other words, if the comparison were made between two other behaviors excluding escapes, will the chronology based parallel pathways still be observed?

- It is known that escapes can inhibit spontaneous swimming by recruiting spinal inhibitory interneurons (Song et al., 2015). This suggests cross talk between the early and late pathways, which according to the authors are parallel and non-overlapping. The authors will do well to discuss this aspect.

- Authors need to discuss the role of hindbrain V2a neurons in struggles. They perform functional imaging during struggles (Figure 4 and Figure 5) but in subsequent experiments, intriguingly, struggles were omitted from the analysis. Struggles are present in early larval stages and are intense but slow and propagate in the opposite direction. Did the authors notice any difference in recruitment of the V2a population that could explain such difference in recruitment of spinal interneurons?

- The authors need to address previous published data including publications from the 3 groups mentioned above, even / especially if the authors do not support the hypothesis that these neurons control speed.

- In invertebrate networks and in mammalian respiratory networks, neuromodulators can reconfigure networks to generate distinct rhythms, patterns and behaviors. This is distinct from the authors' proposal that the emergence of new behaviors relies on the development of entirely new neurons and networks that are added on and are non-overlapping with existing early born networks. Both mechanisms could be operational in different areas of the nervous system but in the current manuscript, the reader gets the opinion that the authors are pushing for the parallel networks idea as an all-encompassing mechanism of network and behavior development. A juxtaposition of these alternate models will benefit the field.

5) The manuscript is already at 105 pages and 11 figures! With the additional analysis required, it will get longer, if some serious attempts at concise writing are not made. The manuscript will benefit from being rewritten more concisely, along with consolidation of figures and the usage of schematics where appropriate.

---

## [Author Response]

We thank the reviewers and the editors for considering our manuscript, and for their insightful and encouraging comments. We have added new analyses and revised the manuscript to address the comments. Please find our responses below.

Essential revisions:The reviewers were unanimous in their praise for the meticulous and thorough investigation undertaken by the authors. However, they also raised a number of concerns that have to be addressed satisfactorily before the paper can be accepted.1) An analysis of hindbrain V2a recruitment with respect to locomotor speeds during escapes, spontaneous swims and struggles. The manuscript covers many aspects of the organization, anatomy and function of V2a hindbrain interneurons without referring ever to locomotor speed or tail beat frequency. Work from the Fetcho, El Manira and McLean labs have previously shown spinal V2a interneurons to be recruited as function of speed. Is there a similar recruitment at the hindbrain level?

We apologize for giving the impression that we neglected the relationship between tail beat frequency and the recruitment of hindbrain V2a neurons. One of the reasons why we focused on these locomotor behaviors in our work was because, these behaviors being kinematically distinct, we could examine the relationship that tail beat frequency and movement strength bear to the recruitment of hindbrain V2a neurons. Now we incorporated an analysis of the tail beat frequency of each locomotor pattern (Figure 4C) and rewrote Results section to make our intent more explicit. We also think that we did not properly explain why we examined struggle-related activity in the original manuscript. Now we introduce struggles in the Introduction section as one of the early-born locomotor patterns that exhibit powerful whole-body bends. We also made more explicit our finding that many of the early-born neurons are active during both escapes and struggles, and that a subset of these neurons shows enhanced activity during escapes (subsection “Hindbrain V2a neurons of different ages are recruited differentially during distinct locomotor patterns”, subsection “Chronological layering of spinal projections of hindbrain V2a neurons suggests parallel descending pathways organized by birthdate”). This observation suggests that the recruitment of the early-born group leads to the strong axial motor activity that entails whole-body bends observed during both escapes and struggles, and that further activation of this group leads to strong locomotor activity with fast tail beat frequency, as observed in escapes. This idea of recruitment based on the strength of movement is also consistent with the connectivity of the early-born population directly innervating primary motoneurons across a large portion of the spinal cord (subsection “Neuronal excitability and synaptic inputs in V2a pathways vary in accordance with birthdate-related recruitment pattern”, subsection “Behavioral contributions of distinct age groups of hindbrain V2a descending neurons”). At first glance, this recruitment pattern may seem at odds with previous studies in the spinal cord that revealed speed-dependent recruitment pattern of V2a neurons **(Kimura, Okamura and Higashijima, 2006; McLean et al., 2008; Ampatzis et al., 2014)**. However, we think this is actually consistent with previous literature as the strength of body bends and tail beat frequency increases together as locomotor speed increases in the context of forward locomotion **(McLean et al., 2008)**. This means that it is possible that spinal V2a neurons, like the hindbrain population, are recruited based on the strength of body bends. Consistent with this idea, earlier-born spinal V2a neurons have longer axons than later-born ones **(Menelaou, VanDunk and McLean, 2014)**, suggesting that the early-born population contributes to large body bends through its innervation of motoneurons across many segments. Indeed, the activity of CiD interneurons, putative early-born V2a population, is enhanced when fish exhibit a larger initial bend during escape **(Bhatt et al., 2007)**. Studies in tadpole also identified dINr, a class of putative spinal V2a neurons, that is recruited during struggles but not during slow swims **(Li et al., 2007)**. This cell type also fires during fast beat frequencies **(Li et al., 2007; Li, 2015)**, indicating that it may correspond to the early-born population active during fast swimming in zebrafish. Taken together, these findings and observations raise the possibility that spinal V2a neurons, similar to the hindbrain population we described, exhibit a recruitment pattern that is based on the strength of movements. We incorporated these points in Discussion section.

Further, escapes in the larva consist of a C bend followed by fast swimming that largely transitions to slow swimming (Mirat et al., 2013; Knafo et al., 2017; Marques et al., 2018). This implies that during the "crude" escape, locomotion starts fast and transitions to slow. How is this transition implemented at the level of the hindbrain V2a interneurons?

This is an interesting point, and one that we were cognizant of at the time of preparing our work for submission. The age-related recruitment pattern we observed in the hindbrain V2a neurons suggests that, during escape, neural activity shifts from the early-born population contributing to the fast cyclic whole-body movements in the early phase of escape to the late-born population contributing to the refined tail-restricted movements in the late phase of escape. Whole-cell recordings of V2a reticulospinal neurons in rhombomere 4 shows that this is indeed the case. This raises the possibility that the recruitment of the early-born and late-born group is coordinated to ensure smooth transition from early-born to late-born neurons. However, we currently do not know the circuit mechanisms underlying this coordination. We included these points in the paragraph discussing the coordination among different age groups (Discussion section).

2) Registration of images against a standard zebrafish brain atlas. The recent efforts made by the community to generate a brain atlas for zebrafish larva (see for example the ZBB from Harry Burgess) is critical here for registering the position of the soma and neuropile of V2a interneurons. In many instances (such as the caudal medial V2a cells), it is not clear where in the dorsoventral axis the soma of V2a cells are located. When presenting specific groups of V2a interneurons, the authors need to register their stacks in the ZBB atlas (or equivalent brain atlas) and add supplemental videos showing where the different V2a types referred to in the manuscript are located. This will also be helpful in Figure 4, where it is not clear how cells were classified as rostral, caudal, dorsal or ventral.

Thank you for the suggestions. We registered the stacks of optical backfill, birthdating photoconversion, and activation maps to the ZBB brain atlas and made supplemental videos (Video 1, Video 2, Video 4 and Video 5) showing the locations of different V2a types. We also incorporated a schema (Figure 2C) to clarify how subgroups of RS neurons are defined based on their rostrocaudal and dorsoventral positions and made it clear in the main text (subsection “Neuronal birth order dictates cell body position and order of spinal projections in hindbrain V2a population”) that this definition is based on previous literature **(Kimmel, Powell and Metcalfe, 1982; Mendelson, 1986).

It is also not clear how the authors ensured that they recorded from the same cells across fish.

Most, if not all, of the early-born neurons are part of dorsal reticulospinal neurons that are individually identifiable based on their characteristic soma position and shape. These neurons served as landmarks for segmental boundaries and helped us identify clusters of ventral RS neurons based on their stereotypical position relative to the dorsal RS neurons (see Video 1). For the early-born V2a neurons in the caudal hindbrain (segment 7/8), we identified them based on the distinct displacement of their soma outside the V2a cluster **(Kinkhabwala et al., 2011)**. We added more detailed descriptions in Results section and Materials and methods section and also provided supplementary videos (Video 1, Video 2, Video 4 and Video 5) and a schema (Figure 2C) to better illustrate their positions.

In addition, for Figure 7 and Figure 8, authors should analyze dorso-ventral position of the spinal targets as a further indicator of whether there is speed-dependent recruitment of the MiV1 cluster.

Here we assume reviewers are asking about the soma position of MiV1 neurons because Figure 7 does not contain information about the spinal targets.

Furthermore, Figure 8E already shows the dorsal extent of the spinal projections of MiV1 age groups. We incorporated measurements of the dorso-ventral positions of the cell bodies (Results section) and clarified the definition of each category in Results section and Materials and methods section.

3) Analysis of behaviors after ablation: The authors analyzed only the initial 6 bends of each swim episode? Why not all of them? Also, what is the result when you look at more global parameters such as total distance traveled and speed for both escape and the 3 spontaneous swim types? Furthermore, did you observe a difference in latency for the escape response? Did you analyze struggles after the ablations?

Our bend-by-bend analysis is based on the assumption that each swim type is so stereotypical that one could compare identically-numbered bends across episodes. In the original manuscript, we made a qualitative judgement that the body bends become less stereotypical after the sixth bend based on the aligned traces of body curvature (Figure 10A vi) and restricted our analysis accordingly to avoid overreaching with our conclusions. Now we assess this assumption more quantitatively based on the timing of each bend using ROC analysis (Figure 10—figure supplement 3; subsection “Behavioral contributions of distinct age groups of hindbrain V2a descending neurons”, subsection “Statistical assessment of the effects of ablations”): we examined how discriminable the time of maximum amplitude of each bend is from that of the preceding bend and found that the discriminability drops substantially after the sixth*^^* bend for escapes and by the tenth*^^* bend for other swim types. However, to better satisfy the reviewers’ request, we moved the upper limit of the bend number to the tenth*^^* bend, which marks the termination of most swim types (Figure 10—figure supplement 2, number of bends per episode). The extension of our analyses to the tenth*^^* bend did not alter the conclusions we had reached in our previous analysis.

Our revised manuscript now also includes analyses of the following parameters: total swim distance per episode, mean swim velocity per episode, maximum swim velocity per episode, number of bends per episode, episode duration, and response latency for escapes (Figure 10—figure supplement 2; subsection” Behavioral contributions of distinct age groups of hindbrain V2a descending neurons”).

We were not able to examine struggle behavior because the gradual mechanical stimulus we used led to highly variable motor patterns across trials in terms of onsets, number, strength of struggle bends. We also lack a high throughput assay to overcome this trial variability. We hope to address this by establishing a well-controlled high throughput assay in the future.

4) Rewriting the Discussion section to include a more comprehensive view of circuit development and emergence of behaviors. There are several issues that have not been adequately addressed by the authors in the Discussion section:- The existence of dedicated interneuronal and motoneuronal classes for mediating escapes is well established in zebrafish through work from multiple labs. Could it be that, escape, being critical for survival matures very early and has dedicated pathways and that the correlation with chronology is incidental? In other words, if the comparison were made between two other behaviors excluding escapes, will the chronology based parallel pathways still be observed?

We made it explicit that the early-born population is shared in both escapes and struggles in Results section and Discussion section to mitigate the impression that the early-born pathway is specific to escapes. We also discussed tadpole studies describing putative spinal V2a neurons that are active during struggling and fast swimming to further emphasize that these two behaviors recruit overlapping sets of neurons (Discussion section).

- It is known that escapes can inhibit spontaneous swimming by recruiting spinal inhibitory interneurons (Song et al., 2015). This suggests cross talk between the early and late pathways, which according to the authors are parallel and non-overlapping. The authors will do well to discuss this aspect.

Incorporated this in Discussion section.

- Authors need to discuss the role of hindbrain V2a neurons in struggles. They perform functional imaging during struggles (Figure 4 and Figure 5) but in subsequent experiments, intriguingly, struggles were omitted from the analysis. Struggles are present in early larval stages and are intense but slow and propagate in the opposite direction. Did the authors notice any difference in recruitment of the V2a population that could explain such difference in recruitment of spinal interneurons?

Described the difference in the recruitment pattern in Results section and also discussed their role in struggles (Discussion section).

- The authors need to address previous published data including publications from the 3 groups mentioned above, even / especially if the authors do not support the hypothesis that these neurons control speed.

Incorporated this in Discussion section.

- In invertebrate networks and in mammalian respiratory networks, neuromodulators can reconfigure networks to generate distinct rhythms, patterns and behaviors. This is distinct from the authors' proposal that the emergence of new behaviors relies on the development of entirely new neurons and networks that are added on and are non-overlapping with existing early born networks. Both mechanisms could be operational in different areas of the nervous system but in the current manuscript, the reader gets the opinion that the authors are pushing for the parallel networks idea as an all-encompassing mechanism of network and behavior development. A juxtaposition of these alternate models will benefit the field.

Incorporated this in Discussion section.

5) The manuscript is already at 105 pages and 11 figures! With the additional analysis required, it will get longer, if some serious attempts at concise writing are not made. The manuscript will benefit from being rewritten more concisely, along with consolidation of figures and the usage of schematics where appropriate.

We made an attempt to make our manuscript more concise, but could not avoid the addition of 6 pieces of supplementary material, which were generated by the new analyses motivated by the reviewers’ requests. Our text, however, spans 4 fewer pages than the original text. We understand our manuscript is still long (83 pages), but our study characterized both the development and function of hindbrain V2a descending neurons, and provided multiple lines of evidence to support our conclusions. We believe all of these results are essential to convincingly support our conclusions. Emboldened by the vision of *eLife*, which encourages scientists to fully describe their results “without any artificial limits on the number of pages, figures or references” **(Schekman, Watt and Weigel, 2012)** and the influential papers** published in *eLife* that took advantage of this policy **(Aso et al., 2014, 25 figures; Aso et al., 2014, 15 figures)**, we would like to ask the editors and reviewers to let us describe the results in this unabridged format.